



# Role of Criegee intermediates in the formation of sulfuric acid at a Mediterranean (Cape Corsica) site under influence of biogenic emissions

Alexandre Kukui[1], Michel Chartier[1], Jinhe Wang[2,3], Hui Chen[3,a], Sébastien Dusanter[4], Stéphane Sauvage[4], Vincent Michoud[4,b], Nadine Locoge[4], Valérie Gros[5], Thierry Bourrianne[6], Karine Sellegri[7], Jean-Marc Pichon[7]

[1] Laboratoire de Physique et Chimie de l'Environnement et de l'Espace (LPC2E), CNRS Orléans, France
[2] Resources and Environment Innovation Research Institute, School of Municipal and Environmental Engineering, Shandong Jianzhu University, Jinan 250101, China
[3] ICARE-CNRS, 1 C Av. de la Recherche Scientifique, 45071 Orléans CEDEX 2, France
[4] IMT Lille Douai, Institut Mines -Télécom, Univ. Lille, Centre for Energy and Environment, F-59000 Lille, France
[5] Laboratoire des Sciences du Climat et de l'Environnement (LSCE), UMR CNRS-CEA-UVSQ, IPSL, Univ. Paris-Saclay, F-91191 Gif Sur Yvette, France
[6] Centre National de Recherches Météorologiques (CNRM), GMEI, MNP, Météo-France, F-31057 Toulouse, France
[7] Laboratoire de Météorologie Physique Observatoire de Physique du Globe (LAMP), Clermont-Ferrand, France
[a] now in Shanghai Key Laboratory of Atmospheric Particle Pollution and Prevention, Department of Environmental Science and Engineering, Institute of Atmospheric Sciences, Fudan University, Shanghai 200438, China
[b] now in Université de Paris, Université Paris Est Créteil (UPEC), UMR CNRS 7583, Laboratoire Interuniversitaire des Systèmes Atmosphériques, Paris, France

*Correspondence to*: Alexandre Kukui (alexandre.kukui@cnrs-orleans.fr)

**Abstract.** Reaction of stabilized Criegee Intermediates (SCIs) with $SO_2$ was proposed as an additional pathway of gaseous sulfuric acid ($H_2SO_4$) formation in the atmosphere, supplementary to the conventional mechanism of $H_2SO_4$ production by oxidation of $SO_2$ in reaction with OH radicals. However, because of a large uncertainty in mechanism and rate coefficients for the atmospheric formation and loss reactions of different SCIs, the importance of this additional source is not well established. In this work, we present an estimation of the role of SCIs in $H_2SO_4$ formation at a western Mediterranean (Cape Corsica) remote site, where a comprehensive field observations including gas phase $H_2SO_4$, OH radicals, $SO_2$, volatile organic compounds (VOCs) and aerosol size distribution measurements have been performed in July - August 2013 as a part of the project ChArMEx. The measurement site was under strong influence of local emissions of biogenic volatile organic compounds including monoterpenes and isoprene generating SCIs in reactions with ozone and, hence, presenting an additional source of $H_2SO_4$ via $SO_2$ oxidation by the SCIs. Assuming the validity of a steady state between $H_2SO_4$ production and its loss by condensation on existing aerosol particles with a unity accommodation coefficient, about 90% of the $H_2SO_4$ formation during the day could be explained by the reaction of $SO_2$ with OH. During the night the oxidation of $SO_2$ by OH radicals was found to contribute only about 10% to the $H_2SO_4$ formation. The accuracy of the derived values for the contribution of OH+$SO_2$ reaction to the $H_2SO_4$ formation is limited mostly by a large, presently of a factor of 2, uncertainty in OH+$SO_2$ reaction rate coefficient. The contribution of the $SO_2$ oxidation by SCIs to the $H_2SO_4$ formation was evaluated using available measurements





of unsaturated VOCs and steady state SCIs concentrations estimated by adopting rate coefficients for SCIs reactions based on structure–activity relationships (SARs). The estimated concentration of the sum of SCIs was in the range of $(1 - 3) \times 10^3$ molecule cm$^{-3}$. During the day the reaction of SCIs with SO$_2$ was found to account for about 10% and during the night for about 40% of the H$_2$SO$_4$ production, closing the H$_2$SO$_4$ budget during the day but leaving unexplained about 50% of the H$_2$SO$_4$

formation during the night. Despite large uncertainties in used kinetic parameters, these results indicate that the SO$_2$ oxidation by SCIs may represent an important H$_2$SO$_4$ source in VOCs-rich environments, especially during night-time.

## 1 Introduction

Sulfuric acid, H$_2$SO$_4$, is an important atmospheric component identified to play a key role in formation of secondary atmospheric aerosol through new particles formation processes (Dunne et al., 2016; Paasonen et al., 2010; Sipilä et al., 2010;

Weber et al., 1997). Also, a noticeable fraction of nucleation mode particle's growth can be explained by sulphuric acid condensation (Boy et al., 2005; Smith et al., 2005). It is therefore important to well understand the atmospheric mechanisms determining the H$_2$SO$_4$ concentrations in different atmospheric environments.

Until recently it was generally accepted that the dominant atmospheric source of H$_2$SO$_4$ is the reaction of OH radicals with SO$_2$ (R1) presenting a rate limiting step leading in the troposphere to a fast production of H$_2$SO$_4$ in presence of water vapour

and oxygen via reactions (R2 - R3) (Finlayson-Pitts and Pitts Jr, 2000). It was assumed that H$_2$SO$_4$ atmospheric concentrations are determined predominantly by this source and the loss of sulphuric acid on the surface of existing particles with the loss rate depending on the efficiency of H$_2$SO$_4$ uptake.

$$OH + SO_2 \rightarrow HSO_3 \quad\quad\quad (R1)$$

$$HSO_3 + O_2 \rightarrow SO_3 + HO_2 \quad\quad\quad (R2)$$

$$SO_3 + H_2O\ (+H_2O) \rightarrow H_2SO_4 + H_2O \quad\quad\quad (R3)$$

Another possible atmospheric source of H$_2$SO$_4$ via oxidation of SO$_2$ by stabilized Criegee intermediates (SCIs), compounds formed by ozonolysis of unsaturated organic compounds, was suggested by Cox and Penkett (1971) and discussed first in view of its atmospheric importance by Calvert and Stockwell (1983). For a long time the reactions of SO$_2$ with SCIs were considered as being too slow to represent an important atmospheric source of H$_2$SO$_4$, until in the more recent study of Welz et al. (2012)

a rate constant of $(3.9\pm0.7)\times10^{-11}$ cm$^{-3}$ molecule$^{-1}$ s$^{-1}$ was derived for the reaction of SO$_2$ with the simplest SCI, formaldehyde oxide (CH$_2$OO), which is significantly larger than previous estimates of around $4\times10^{-15}$ cm$^{-3}$ molecule$^{-1}$ s$^{-1}$ for the reactions of SO$_2$ with CH$_2$OO (Hatakeyama et al., 1986) and for the reactions of SO$_2$ with other SCIs (Johnson and Marston, 2008). The importance of this additional source of H$_2$SO$_4$ which is still under discussion depends on the atmospheric SCIs concentrations and the kinetics and mechanisms of the SCIs reactions with SO$_2$.

Criegee intermediates (CIs), also known as carbonyl oxides (R$_1$)(R$_2$)COO with R$_1$ and R$_2$ representing different substituents, are produced via ozonolysis of alkenes by cycloaddition of an ozone molecule on a double bond forming a primary





ozonide (POZ), a highly energized compound containing an O-O-O group. Subsequent rapid cleavage of either of the O-O bonds leads to the formation of chemically activated CIs, which can undergo either prompt dissociation or thermal stabilization leading to the formation of the SCIs (Criegee, 1975; Criegee and Wenner, 1949; Donahue et al., 2011; Johnson and Marston, 2008; Vereecken and Francisco, 2012; Vereecken et al., 2012).

Atmospheric concentrations of SCIs depend on their production rates, by the ozonolysis of alkenes, and their loss rates, predominantly via unimolecular decomposition and reactions with water monomers and dimers (Vereecken et al., 2017). However, the kinetic parameters and the reaction mechanism of these processes are not well known.

The SCIs production rate is determined by the ozonolysis reaction rate constants and corresponding speciated yields of different SCIs. The speciated SCI yield depends on a relative yield of two different CIs formed by decomposition of the POZ and a yield of SCI produced by the CI collisional stabilization depending on the CI structure and its energy content. For most of atmospherically relevant alkenes, the total SCI yields were not studied directly, while for those for which multiple studies are available there is in many cases a large data scatter. The data on the speciated SCI yields are available only for a few of alkenes (see e.g. Vereecken et al., 2017 and references therein).

For the loss of SCIs via monomolecular decomposition and reactions with water monomers and dimers, the results of experimental and theoretical studies show that the corresponding rate coefficients may vary by orders of magnitude depending on the SCI substituents and conformers (see e.g. Vereecken et al., 2017 and references therein). In recent years experimental studies of these reactions with direct detection and generation of specific SCIs were performed for several among the simplest of them, such as formaldehyde oxide ($CH_2OO$) (Chao et al., 2015; Lewis et al., 2015; Sheps et al., 2017; Smith et al., 2015; Stone et al., 2018), acetaldehyde oxide ($CH_3CHOO$) (Li et al., 2020; Lin et al., 2016; Sheps et al., 2014) and acetone oxide (($CH_3)_2COO$) (Chhantyal-Pun et al., 2017; Fang et al., 2017; Huang et al., 2015; Lester and Klippenstein, 2018; Smith et al., 2016). Very recently the decomposition rate and an estimation of the rate coefficients for the reaction with water vapor were obtained for the first time in direct kinetic studies for *syn* Methyl Vinyl Ketone oxide (*syn*-MVK-oxide), a four-carbon unsaturated Criegee intermediate derived from the ozonolysis of isoprene (Barber et al., 2018; Caravan et al., 2020). For other large SCIs only estimations based on theoretical and indirect studies are available for their reactions with water vapor and their thermal decomposition.

The result of Welz et al. (2012) about the fast reaction of $SO_2$ with formaldehyde oxide was extended in later direct and indirect studies where similarly fast reactions with $SO_2$ were confirmed for $CH_3CHOO$, ($CH_3)_2COO$, Z-nopinone oxide (product of β-pinene ozonolysis) and *syn*-MVK-oxide (Ahrens et al., 2014; Caravan et al., 2020; Vereecken et al., 2017 and references therein), with rate coefficients in the range of $(3\text{-}16) \times 10^{-11}$ $cm^3$ molecule$^{-1}$ s$^{-1}$. Reactions of other SCIs with $SO_2$ were suggested to be similarly fast on the basis of theoretical results (Kurtén et al., 2011).

Theoretical analysis suggests that the reaction of SCI with $SO_2$ proceeds via a barrierless cycloaddition of $SO_2$ to SCI forming a sulfur-bearing secondary ozonide (SOZ) which can either be stabilized or decompose to form $SO_3$ or other products (Kurtén et al., 2011; Kuwata et al., 2015; Vereecken et al., 2012). For smaller SCIs, e.g. formaldehyde and acetone oxides, the theory predicts negligible SOZ stabilization and about unity yield of $SO_3$ (Kuwata et al., 2015). These results are supported by



experimental studies for $CH_2OO$ (Berndt et al., 2014a; Wang et al., 2018) and $(CH_3)_2OO$, $CH_3CHOO$ (Berndt et al., 2014b). For larger SCIs with expected longer SOZ lifetime the $SO_3$ yield may depend on the SOZ fate in the atmosphere with respect to its decomposition or further reactions, e.g. with $H_2O$, (Kuwata et al., 2015; Vereecken et al., 2012).

Estimations based on the available or evaluated kinetic parameters show that the atmospheric SCI concentrations vary by
orders of magnitude depending on conditions specific for different environments, such as the concentrations and composition of alkenes, ozone concentration or humidity. Using chemistry-transport global modeling the highest SCI concentrations of the order of $10^4 – 10^5$ molecule $cm^{-3}$ were inferred for the regions with highest isoprene and terpenes emissions, e.g. above the tropical forest (Chhantyal-Pun et al., 2019; Khan et al., 2018; Newland et al., 2018; Vereecken et al., 2017). Estimated using steady-state calculations, the concentration of SCI ranges from $2.3 \times 10^3$ molecule $cm^{-3}$ at a rural site to $5.5 \times 10^4$ molecule $cm^{-3}$
in an urban polluted environment (Vereecken et al., 2017). The estimated contribution of SCI to $H_2SO_4$ formation is also highly variable: about 7% in rural environments and up to 70% over tropical regions (Vereecken et al., 2017). At the global scale, the contribution of SCI to $SO_2$ oxidation was estimated to be negligible, contributing less than 1% (Newland et al., 2018). The uncertainty associated to the predicted SCI concentrations was estimated to be one order of magnitude (Vereecken et al., 2017), due to poorly defined SCI formation and loss rates. Even a higher uncertainty may be expected for the estimated contribution
of SCIs to $SO_2$ oxidation considering not well defined reaction rate coefficients for the reaction of different SCIs with $SO_2$.

The adequacy of the mechanism treating the $SO_2$ oxidation by OH as a predominant source of the atmospheric $H_2SO_4$ was tested in a number of field campaigns where simultaneous measurements of OH and $H_2SO_4$ were conducted (Supplement Table S1). Selected ion Chemical Ionisation Mass Spectrometry (CIMS) technique for simultaneous measurements of OH and $H_2SO_4$ was first introduced by (Eisele and Tanner, 1993) and since then it has been used in a number of field measurements in different
environments. In these studies the measurements of OH, $H_2SO_4$, $SO_2$ and aerosol surface area were used to compare the rate of $H_2SO_4$ production in reaction (R1) and the rate of $H_2SO_4$ loss on aerosol particles assuming steady state condition between these processes. In a number of measurements campaigns the $H_2SO_4$ budget was found to be closed using the uptake coefficient of unity corresponding to upper limit of the $H_2SO_4$ loss rate on existing particles. This was observed in different environments including remote marine (Weber et al., 1997), forested rural (Birmili et al., 2000; Boy et al., 2013) and forested remote sites
(Eisele and Tanner, 1993; Weber et al., 1997). However, in other field studies conducted in various environments the $H_2SO_4$ condensation sink calculated using an uptake coefficient of unity was found to significantly exceed its formation rate via $SO_2$ oxidation by OH indicating either an $H_2SO_4$ uptake efficiency lower than unity or the presence of sources of $H_2SO_4$ other than reaction (R1) (Bardouki et al., 2003; Berresheim et al., 2002, 2014; Boy et al., 2013; Jefferson et al., 1998; Mauldin III et al., 2012; Petäjä et al., 2009). Several additional $H_2SO_4$ gas phase sources were suggested such as the oxidation of DMS or DMDS
in remote coastal environments proceeding with $SO_3$ formation (Berresheim et al., 2002, 2014; Jefferson et al., 1998) or $SO_2$ oxidation by SCIs in the boreal forest and in moderately polluted environments (Boy et al., 2013; Kim et al., 2015; Mauldin III et al., 2012). A heterogeneous formation of gas phase $H_2SO_4$ via the catalytic oxidation of $SO_2$ on the surface of black carbon aerosols has also been recently shown to be important under polluted conditions (Yao et al., 2020).



In this work, we present an evaluation of the role of SCIs in $H_2SO_4$ production at a remote site on Cape Corsica near the North tip of Corsica Island (Ersa station, western Mediterranean). In July - early August 2013, comprehensive field observations including gas phase (OH radicals, $H_2SO_4$, VOCs, $NO_x$, $SO_2$, others) and aerosol size distribution measurements were conducted at this site in the frame of the SAFMED (Secondary Aerosol Formation in the Mediterranean) campaign as part of the summer 2013 experimental effort of the project ChArMEx (the Chemistry-Aerosols Mediterranean Experiment). During the field campaign, the site was strongly influenced by local emissions of biogenic volatile organic compounds, including isoprene and terpenes, forming different SCIs in reactions with ozone and, hence, potentially representing an additional source of $H_2SO_4$ via reactions of SCIs with $SO_2$. We use the OH, $H_2SO_4$ and $SO_2$ measurements to estimate an upper limit for the contribution of $H_2SO_4$ sources other than reaction (R1). Using available measurements of unsaturated VOCs and adopting rate coefficients for SCIs reactions based on structure–activity relationships (SARs) from Vereecken et al. (2017), we estimate steady-state SCIs concentrations. These SCIs concentrations are used for the estimation of the rate of $H_2SO_4$ formation in the reactions of SCIs with $SO_2$ and its comparison with OH+$SO_2$ source resulting from the OH and $H_2SO_4$ measurements.

## 2 Methods

### 2.1 Field site

Measurements were performed at the Ersa site from 18 July to 5 August 2013 during ChArMEx/SAFFMED field campaign (Dulac et al., 2021). The Ersa station (42.969°N, 9.380°E) is located at Cape Corsica on the northern edge of Corsica (Michoud et al., 2017; Zannoni et al., 2017). It is situated at an altitude of 533 meters above sea level on the top of a hill dominating the northern part of the cape. On its eastern, northern and western sides it is a few km away from the coast and has a direct view of the sea. The measurement site is isolated by a mountain range from the closest large city, Bastia, situated about 30 km south of the site. The site is surrounded by widespread vegetation such as scrubland typical of the Mediterranean areas, responsible of biogenic VOC emissions (Debevec et al., 2021; Zannoni et al., 2015).

### 2.2 Experimental Methods

### 2.2.1 OH and $H_2SO_4$ measurements

Concentrations of OH radicals and $H_2SO_4$, as well as total peroxy radicals ($HO_2$+$RO_2$, not discussed here), were measured using chemical ionization mass spectrometry (CIMS) (Berresheim et al., 2000; Eisele and Tanner, 1991). A detailed description of the instrument is presented elsewhere (Kukui et al., 2008, 2012). Here we briefly present the measurement technique and essential details about the setup and performance of the instrument during the ChArMEx/SAFMED campaign. A detailed description of the calibration system used during the campaign, which was not presented before, is given in Supplement Sect. S3.





OH was detected by conversion of the sampled OH with isotopically labelled $^{34}SO_2$ to form $H_2^{34}SO_4$ in a chemical
conversion reactor (CCR) in presence of ambient water vapour and oxygen. The isotopically labeled $H_2^{34}SO_4$ and ambient
$H_2SO_4$ were detected by mass spectrometry as $H^{34}SO_4^-$ and $H^{32}SO_4^-$ product ions. The product ions were produced by chemical
ionization with $NO_3^-$ reagent ion in an ion-molecule reactor (IMR) following the CCR. The reagent ions were generated in a
corona $NO_2$/air discharge ion source (CD). A scheme of the reactor is presented in Supplement, Fig. S2.

Ambient air was sampled at a volumetric flow rate of 10 SLM (Standard Liter per Minute) creating turbulent flow in the
chemical conversion region of the reactor. The turbulent flow conditions minimize possible influence of wind speed on the
measurements and ensure fast mixing of reactants. The reactants used for the chemical conversion ($^{34}SO_2$ for OH conversion
into sulfuric acid and NO for peroxy radicals conversion into OH and their subsequent detection as OH) and the radical
quencher ($NO_2$) are introduced into the reactor through a set of injectors. $NO_2$ used as a scavenger removes not only the OH
radicals, but also peroxy radicals converting them into $HO_2NO_2$ and $RO_2NO_2$ peroxy nitrates. Switching the reactant flows
between the different injectors allows measurements in four different modes: the background mode, two different OH radical
measurement modes and the $RO_2$ radical measurement mode (Supplement Fig. S1). The two OH measurement modes differ
by the time used for the chemical conversion, 4 ms and 20 ms. Ratio of the signals with the short and the long conversion times
may be used as an indicator of an artificial OH formation in the reactor (Kukui et al., 2008).

Measurements were performed by monitoring the peak intensities at m/z=62 ($NO_3^-$), m/z=97 ($H^{32}SO_4^-$), and m/z=99
($H^{34}SO_4^-$) with the CIMS, respectively noted $I_{62}$, $I_{97}$, and $I_{99}$ hereafter. Every measurement of OH was derived from 1 min of
OH ion signal count and two 30 s background ion signal counts before and after the OH signal measurement. $RO_2$ was measured
at the end of the OH detection sequence by switching on the NO flow to the corresponding injector for a duration of 2 min. To
avoid any possible influence of traces of NO on the OH measurements a time delay of 6 min was imposed after switching off
the NO flow and before starting the next OH measurement sequence in order to ensure flushing of the CCR. The OH and the
$H_2SO_4$ data were averaged resulting either in a sequence of 3 points with a step of 7 min separated by a time gap of 15 min or
yielding a sequence with a time step of about 90 min. The latter was used to match the time resolution of the VOCs
measurements (Supplement Table S4).

The concentration of OH and $H_2SO_4$ were derived from the measured ratios of the $H^{32}SO_4^-$, $H^{34}SO_4^-$ and $NO_3^-$ ion peak
intensities, $I_{97}/I_{62}$ and $I_{99}/I_{62}$: $[R] = C_R \times I^R$, where R corresponds to OH or $H_2SO_4$, $I^R$ is a combination of $I_{97}/I_{62}$ and $I_{99}/I_{62}$ ratios
corresponding to OH or $H_2SO_4$ accounting for isotopic composition of $SO_2$ used for the chemical conversion (99% isotopic
enrichment of $^{34}S$, Eurisotop, Cambridge Isotope Laboratories, Inc.) and sulphur isotope natural abundance ($^{32}S$ (95.02%) and
$^{34}S$ (4.21%) (Hoefs, 2018)), $C_R$ is a calibration coefficient determined in calibration measurements by production of OH or
$H_2SO_4$ in a turbulent flow reactor using photolysis of water vapour at 184.9 nm and quantified by chemical actinometry using
photolysis of $N_2O$ (Faloona et al., 2004). A detailed description of the calibration system with definitions of $C_R$ and $I^R$ are
given in Supplement Sect. S3.

The overall accuracy of the calibration coefficients was estimated taking into account uncertainties of all parameters used
for calculation of the radical concentrations in the photolysis reactor and the precision of the measurements of the ratios $I_{97}/I_{62}$



and $I_{99}/I_{62}$. The overall estimated calibration uncertainty ($1\sigma$) was of 30% for OH, 32% for $H_2SO_4$ and 8% for a ratio of $[H_2SO_4]$ to $[OH]$, (Supplement Table S3). Accounting for the calibration uncertainties and the measurements precision, the overall $1\sigma$ uncertainty of the 14 min averaged measurements of OH, $H_2SO_4$ and the ratio of $[H_2SO_4]/[OH]$ was estimated to be around 32%, 34% and 16% during the daytime and 42%, 44% and 40% during the night time, respectively. During the ChArMEx/SAFMED campaign the observed level of OH background signal was significantly higher than typical OH background found during calibration or field measurements in air with low VOCs concentrations (see Fig. 7 and discussion in Sect. 4.4). Accordingly, the lower limits of detection for OH and $H_2SO_4$ at signal-to-noise ratio of 2 and a 15 min integration time were $5\times10^5$ molecule cm$^{-3}$ and $2\times10^5$ molecule cm$^{-3}$, respectively.

During the ChArMEx/SAFMED campaign the instrument was installed in a dedicated container with the CCR fixed to the roof of the container via an interface cap covered with a PTFE sheet. The sampling aperture of the reactor (3 mm diameter) was positioned 50 cm above the roof and about 3 m above the ground.

To avoid possible contamination of ambient air by the $SO_2$, NO and $NO_2$ reactants added to the CCR, a trap was set up at the pumps exhaust by using two 100 L cylinders containing zeolites. The cylinders were refilled several times during measurements. A flexible exhaust tube of 30 m length was always placed downwind from the container.

### 2.2.2 Complementary measurements

The aerosol particle size distribution was measured using a scanning mobility particle sizer (SMPS TSI 3080, associated with a CPC TSI 3010) in the range from 10.9 nm to 495.8 nm and with an aerodynamic particle sizer (APS, TSI 3321) in the range from 542 nm to 19.48 µm. As the SMPS measurements were made with dehydrated particles the particle diameters were corrected to ambient humidity using particle growth-factor (GF) of 1.5 at 90% determined with Volatility Hygroscopic—Tandem Differential Mobility Analyzer (VH-TDMA) (Villani et al., 2008). The dependence of the GF on relative humidity (RH) was calculated using the one-parameter approximation from Rissler et al. (2006). An estimated uncertainty of measured particle number densities and GF corrected particle diameters were of 10% and 15%, respectively.

$SO_2$ concentration was measured by UV fluorescence (Thermo Environmental Instruments (TEI), Model 43C-TLE) with an estimated accuracy of 20%, a lower detection limit of 0.05 ppb and a time resolution of 5 min.

Ozone concentration was measured by means of a CraNOx II (Eco Physics) NOx and $O_3$ monitor with an estimated accuracy of 10%.

A detailed description of VOCs measurements during ChArMEx/SAFMED campaign is given in Michoud et al. (2017). The measurements of 23 unsaturated VOCs including alkenes, aldehydes, ketones, isoprene and monoterpenes were used in this work for estimation of SCIs concentrations. Employed measurement techniques and concentration ranges for measured unsaturated VOCs are given in Supplement Table S4 together with associated time resolution, limit of detection and uncertainties. The data were averaged or interpolated with a time step of 90 min.

Wind speed and direction, relative humidity, temperature and photolysis rates were also measured throughout the campaign.



### 2.3 Estimation of $H_2SO_4$ steady state concentrations

Concentrations of $H_2SO_4$ produced via $SO_2$ oxidation by OH and sum of SCIs, $H_2SO_4^{OH}$ and $H_2SO_4^{SCI}$, respectively, were calculated assuming validity of a steady state between the $H_2SO_4$ production and its loss (see discussion in Sect. 4.1),

$$\left[H_2SO_4\right]^{OH} = \frac{k_1 \cdot \left[OH\right] \cdot \left[SO_2\right]}{CS} \tag{1}$$

$$\left[H_2SO_4\right]^{SCI} = \frac{\left\{\sum_i k_{SO2}^i \cdot \left[SCI^i\right]\right\} \times \left[SO_2\right]}{CS} \tag{2}$$

Here [OH] and $[SO_2]$ are measured concentrations, $[SCI^i]$ are estimated concentrations of speciated SCIs and $CS$, a condensation sink, is a rate of $H_2SO_4$ loss by its condensation on aerosol particles which is assumed to be the predominant $H_2SO_4$ loss process, while the dry deposition on macroscopic surfaces is neglected considering a long associated life time estimated of about 1 day using a typical deposition velocity of 1 cm s$^{-1}$ (Seinfeld and Pandis, 2016) and boundary layer depth of 1 km. Considering the $CS$ as a major sulfuric acid loss process the median value of $H_2SO_4$ lifetime was 2.7 min (2.2 min – 3.7 min interquartile range, Fig. 1, 2). The rate coefficient for the reaction of OH with $SO_2$, $k_1$=8.06×10$^{-13}$ cm$^3$ molecule$^{-1}$ s$^{-1}$ (at 760 torr and 298 K), was taken from IUPAC 2004 recommendation (Atkinson et al., 2004) (see Discussion Sect.). Production of $H_2SO_4^{SCI}$ is calculated as a sum of contributions from different speciated $SCI^i$ reacting with $SO_2$ with rate coefficients $k_{SO_2}^i$ (Supplement Table S5). It is assumed here that the $H_2SO_4$ yield in reaction of SCIs with $SO_2$ is a unity for all SCIs, giving an upper limit for the contribution to $H_2SO_4$ formation of accounted SCIs.

The $CS$ was calculated using measured particle size distributions and number concentrations by calculating diffusional flux to the aerosol particles assuming an accommodation coefficient of a unity (Hanson, 2005) and using Fuchs-Sutugin transition correction (Jefferson et al., 1998; Seinfeld and Pandis, 2016). Diffusion coefficient of $H_2SO_4$ was estimated using its dependence on relative humidity from (Hanson and Eisele, 2000) giving 0.077 cm$^2$ s$^{-1}$ at RH of 60.6% (median value). Without considering an uncertainty on the accommodation coefficient, the accuracy of the calculated $CS$ of 20% was assessed accounting for the uncertainties of particle measurements given in Sect. 2.2.2 and an uncertainty of $H_2SO_4$ diffusion coefficient of 5%.

Concentrations of speciated SCIs, $[SCI^i]$, were calculated assuming steady state conditions considering their production by ozonolysis of the measured unsaturated VOCs and their loss by thermal decomposition and in reactions with water vapour (with $H_2O$ and $(H_2O)_2$):

$$\left[SCI^i\right] = \frac{\left\{\sum_X k_{X+O_3} \cdot \alpha_X^i \cdot Y_X^i \cdot \left[X\right]\right\} \times \left[O_3\right]}{K^i + k_{H_2O}^i \cdot \left[H_2O\right] + k_{(H_2O)_2}^i \cdot \left[(H_2O)_2\right]} \tag{3}$$





where X denotes a specific VOC, $k_{X+O_3}$ is a rate coefficient for the reaction of X with $O_3$, $\alpha_x^i$ is a yield of stabilized $SCI^i$ from

$CI^i$, $Y_x^i$ is a specific yield of $CI^i$ in the reaction X+$O_3$, and $K^i$, $k_{H_2O}^i$, $k_{(H_2O)_2}^i$ are rate coefficients for the thermal decomposition,

the reaction with $H_2O$ and the reaction with water dimer for the specific $SCI^i$, respectively.

The rate coefficients and the yields $\alpha_x^i$ and $Y_x^i$ for 36 SCIs derived from the ozonolysis of 23 measured VOCs are presented

in Supplement Tables S4 and S5. Apart from a few rate coefficients and yields available from more recent experimental and

theoretical studies, most of the parameters in Eq. (2) and Eq. (3) are taken from recommendations of Vereecken et al. (2017),

which are grounded either on an analysis of available experimental data or derived from a theory based structure–activity

relationships (SARs). For the 36 SCIs listed in Tables S4 and S5, we have adopted the SCIs naming convention from Vereecken

et al. (2017). Uncertainty on the estimated in this way SCI concentrations is estimated in Vereecken et al. (2017) to be of an

order of magnitude, mainly due to the uncertainties in CI speciation and the SCI decay rates.

## 3 Results

### 3.1 Observed data

      Time series and median diel profiles of observed concentrations of OH, $H_2SO_4$, $SO_2$, $O_3$, selected VOCs and calculated

condensation sink of $H_2SO_4$ are presented in Fig. 1 (time series) and Fig. 2 (median diel cycles, also including $J(O^1D)$, $T$, $RH$,

and $CS$ data). The night time OH and $H_2SO_4$ measurements on the 26 and 28-30 of July were influenced by strong fog event

deteriorating the accuracy of the corresponding OH and $H_2SO_4$ data.

      OH and $H_2SO_4$ concentrations reached around midday the maximum values of $4.3 \times 10^6$ molecule $cm^{-3}$ and $8.5 \times 10^6$ molecule

$cm^{-3}$ (mean values for day time hours from 11:00 to 13:00 local time (= GMT + 2 h)), respectively, with night-time

concentrations around $1 \times 10^5$ molecule $cm^{-3}$ and $5 \times 10^5$ molecule $cm^{-3}$ for OH and $H_2SO_4$, respectively, close to the detection

limit.

      Comparing the present observations (Fig. 2) with previous measurements in the Mediterranean region, the mean peak OH

concentration during the noon hours was close to the peak OH levels observed during CYPHEX campaign in the summer of

2014 in Cyprus in the eastern Mediterranean, $5.8 \times 10^6$ molecule $cm^{-3}$ (Mallik et al., 2018), where $O_3$ concentrations and $J(O^1D)$

peak levels were similar to those observed during ChArMEx ($J(O^1D)$ was measured during ChArMEx, but not yet published).

The somewhat lower ChArMEx OH noon concentrations compared to CYPHEX are consistent with higher OH reactivity

observed during ChArMEx (Zannoni et al., 2017), although the direct comparison of radical chemistry at these two sites is not

straightforward considering that, among other differences between these sites, biogenic VOCs at Cyprus site, e.g. isoprene and

monoterpenes, were 3-5 times lower compared to ChArMEx. About four times higher OH peak concentrations, $2.1 \times 10^7$

molecule $cm^{-3}$, were observed during the MINOS campaign in the summer of 2001 in Crete (central Mediterranean) with



similar $O_3$ and $J(O^1D)$ observed levels (Berresheim et al., 2003). This difference is difficult to explain based on the available data (Mallik et al., 2018).

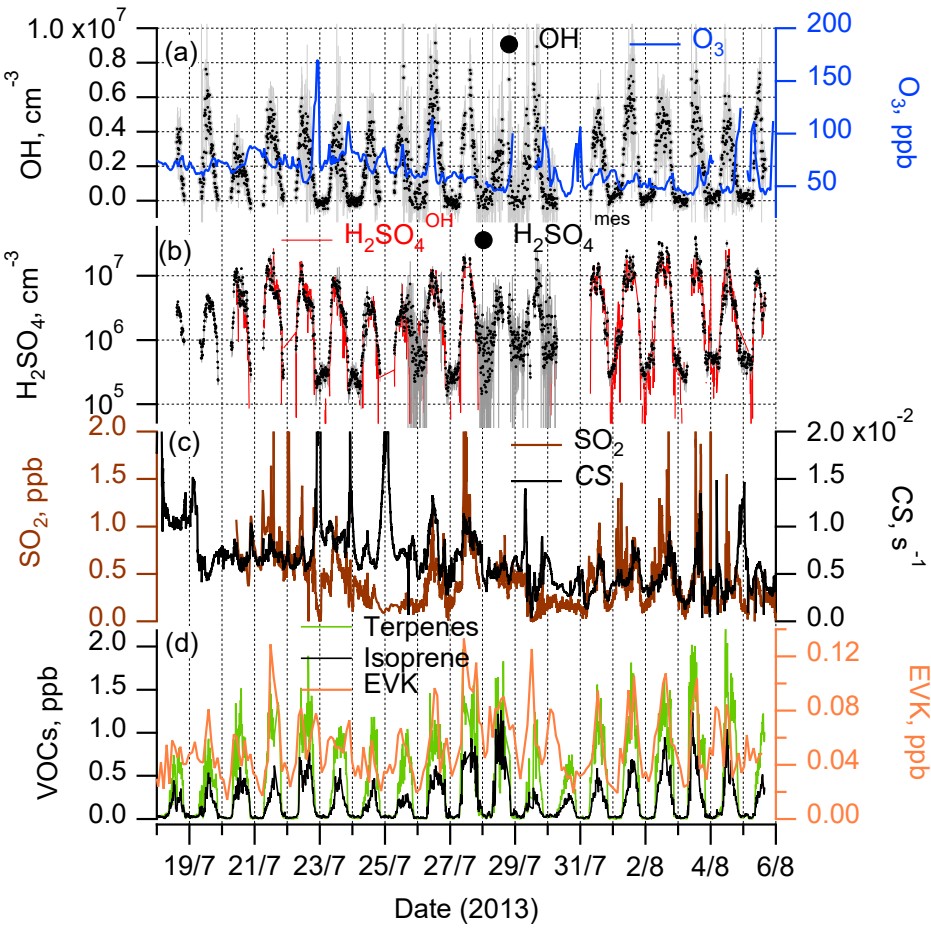

**Figure 1**. Time series of the observations during the ChArMEx summer 2013 campaign: OH radicals and $O_3$ (a); sulfuric acid observed, $H_2SO_4^{mes}$, and calculated assuming only $SO_2$+OH source, $H_2SO_4^{OH}$ (see Sect. 2.3) (b); $SO_2$ and condensation sink $CS$ (c); total monoterpenes and isoprene (left axis), EVK (right axis) (d).

The observed ChArMEx $H_2SO_4$ concentrations were about two times higher than the observed OH concentrations. For other sites the observed ratios $[H_2SO_4]/[OH]$ were in the range from 1 to 9 with only one example when this ratio was less than unity (Supplement Table S1). The ratio $[H_2SO_4]/[OH]$ depends on the $SO_2$ concentration and condensation sink correlating with the aerosol particle surface area concentration. The condensation sink and $[SO_2]$ values during ChArMEx, with median values of $6.1\times10^{-3}$ $s^{-1}$ and 0.44 ppb, respectively, were typical for clean remote continental or coastal environments (Supplement Table S1).



**Figure 2**. Median diel profiles of calculated [$H_2SO_4$] produced in $OH+SO_2$ ($H_2SO_4^{OH}$), observed [$H_2SO_4$], [OH] and photolysis rate $J(O^1D)$ **(a)**, condensation sink (CS) and [$SO_2$] **(b)**, [$O_3$] and concentrations of selected VOCs **(c)**, relative humidity (*RH*)

and temperature (*T*) **(d)**. Shaded areas represent the 25/75 percentiles.

The unsaturated VOCs observed during ChArMEx showed strong diel variation from about 0.4 ppb during the night to about 2 ppb at noon for the concentration of the sum of the VOCs (Fig. 1, 2, Supplement Fig. S9, Table S4). The major contribution during the day (from 7:00 to 20:00) was from biogenic VOCs (isoprene (22% on average), β-pinene (14%), α-

pinene (9%), α-terpinene (7%), methyl vinyl ketone (MVK) (5%) and methacrolein (MACR) (5%)) with emission rates correlating with temperature and solar radiation (Kesselmeier and Staudt, 1999). At night-time, the unsaturated VOCs were represented mostly by compounds of mixed biogenic and anthropogenic origin: ethene (31% on average), acrolein (14%) and





ethyl vinyl ketone (9%). A significant night time contribution was also found from isoprene (9%) and its first-generation oxidation products MVK (6%) and MACR (6%).

**3.2 Comparison of observed H₂SO₄ with sulfuric acid produced from OH+SO₂ (H₂SO₄$^{OH}$)**

Time series of the $H_2SO_4$ produced in the reaction of OH with $SO_2$, $H_2SO_4^{OH}$, calculated according to Eq. (1) and of the observed $H_2SO_4$ are presented in Fig. 1b and 2b showing apparently good agreement between them during the day and an underestimation of the sulfuric acid concentration by $H_2SO_4^{OH}$ during the night. The inverse variance weighted mean value of the ratio of $[H_2SO_4]^{OH}/[H_2SO_4]$ for the data presented in Fig. 1b is $0.86 \pm 0.04$ during the day (7:00 – 20:00) and $0.09 \pm 0.02$ during the night (20:00 – 7:00), respectively (Table 1). Uncertainties on the ratios $[H_2SO_4]^{OH}/[H_2SO_4]$ were estimated by accounting for the uncertainty on the measured ratio of $[OH]/[H_2SO_4]$ (see Sect. 2.2.1), on the $SO_2$ measurements (20%) and on the $CS$ calculations (20%), without considering uncertainties on the reaction rate coefficient $k_1$ and the $H_2SO_4$ uptake coefficient (see Sect. 4.2 and Sect. 4.3).

**Table 1.** Comparison of observed $[H_2SO_4]$ with calculations assuming $H_2SO_4$ formation via oxidation of $SO_2$ by OH and SCIs. [OH] and $[H_2SO_4]$ are observed concentrations, $[H_2SO_4]^{OH}$ and $[H_2SO_4]^{SCI}$ are calculated $H_2SO_4$ produced by oxidation of $SO_2$ by OH and SCI, respectively.

| | Daytime: 7:00 – 20:00 | | Night-time: 20:00 – 7:00 | |
|---|---|---|---|---|
| | Median (inter-quartile range) | Mean ± 1σ | Median (inter-quartile range) | Mean ± 1σ |
| [OH], $10^5$ cm$^{-3}$ | 31 (18; 42) | 31 ± 17 | 1.1 (-0.7; 3.0) | 1.7 ± 4.0 |
| $[H_2SO_4]$, $10^5$ cm$^{-3}$ | 47 (28; 86) | 63 ± 49 | 4.2 (3.1; 6.4) | 5.8 ± 4.8 |
| | | | | |
| $[H_2SO_4]^{OH} = a + b \times [H_2SO_4]$ | $a=(-2.0 \pm 0.5) \times 10^5$ ; $b=0.85 \pm 0.02$ | | $a=(-3.1 \pm 0.3) \times 10^5$ ; $b=0.97 \pm 0.1$ | |
| | | | | |
| $[H_2SO_4]^{OH}/[H_2SO_4]$, % | 95 (79; 129) | 86 ± 4 | 39 (-8; 84) | 9 ± 2 |
| | | | | |
| $1-[H_2SO_4]^{OH}/[H_2SO_4]$, % | 5 (-29; 21) | 14 ± 4 | 61 (16; 108) | 91 ± 2 |
| $[H_2SO_4]^{SCI}/[H_2SO_4]$, % | 10 (7; 16) | 12 ± 6 | 30 (22; 48) | 38 ± 24 |
| | | | | |
| $[H_2SO_4]-[H_2SO_4]^{OH}$, $10^5$ cm$^{-3}$ | 1.2 (-12.5; 8.1) | 4.6 ± 3.2 | 3.0 (0.8; 5.2) | 3.1 ± 0.4 |
| $[H_2SO_4]^{SCI}$, $10^5$ cm$^{-3}$ | 6.0 (3.7; 8.6) | 6.4 ± 3.7 | 1.4 (1.1; 2.4) | 1.8 ± 1.2 |

Considering the correlation of the calculated sulfuric acid concentration $[H_2SO_4]^{OH}$ with the measured $[H_2SO_4]$ shown in Fig. 3, we find that the linear regression using a bivariate fit procedure accounting for the measurement errors of both $H_2SO_4$ and $H_2SO_4^{OH}$ (York et al., 2004) results in slopes of $0.85 \pm 0.02$ and $0.97 \pm 0.1$ for the daytime and the night-time, respectively. The linear regression yields significant intercepts of $(-2.0 \pm 0.5) \times 10^5$ molecule cm$^{-3}$ during the day and of $(-3.1 \pm 0.3) \times 10^5$ molecule cm$^{-3}$ at night. The close to unity slopes and the negative intercepts may be interpreted as a presence of an additional source contributing to $H_2SO_4$ on an average level of several of $10^5$ molecule cm$^{-3}$. During the night this source is important comparing



with the observed night-time average $[H_2SO_4]$ of $(5.8 \pm 4.8) \times 10^5$ molecule cm$^{-3}$. During the night the estimation of the contribution from missing sources derived from the linear fit is in qualitative agreements with the mean ratio of $[H_2SO_4]^{OH}/$ $[H_2SO_4]$ of 0.09 indicating that the $SO_2+OH$ source explains only about 10% of the $H_2SO_4$ formation. During day time, the reaction of $SO_2$ with OH source explains around 90% of the $H_2SO_4$ formation (Table 1).

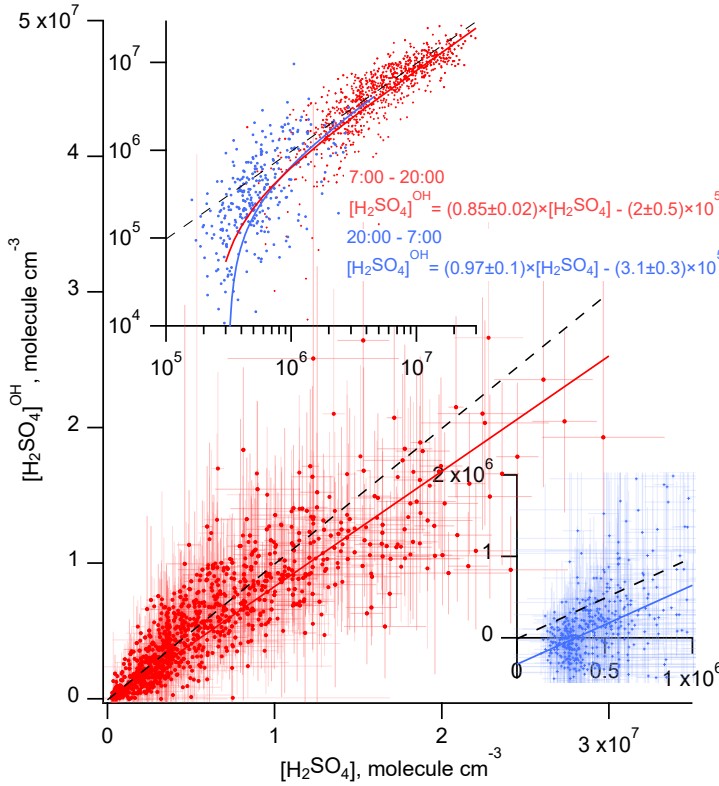

**Figure 3**. Comparison of the measured $H_2SO_4$ and the $H_2SO_4^{OH}$ calculated accounting only for the OH+SO$_2$ source during the day (red) and during the night (blue). Solid lines correspond to linear regression fitting accounting for both X and Y measurement uncertainties. Dashed lines represent 1:1 ratios.

### 3.3 Comparison of observed H$_2$SO$_4$ with sulfuric acid produced from SCIs+SO$_2$ (H$_2$SO$_4^{SCI}$)

Estimated according to Eq. (2) and (3), mean diel profiles of the sulphuric acid $H_2SO_4^{SCI}$ produced in the reactions of $SO_2$ with SCIs generated by the ozonolysis of the measured unsaturated VOCs, excluding α-terpinene, are presented in Fig. 4a. The calculated contribution from α-terpinene alone is up to six times larger than observed $[H_2SO_4]$ (Fig. 4b). It is not clear if the reason for this large overestimation can be related to an erroneous α-terpinene measurements and/or to incorrect kinetic parameters used for the calculation of $H_2SO_4$ production from α-terpinene. The observed α-terpinene daytime concentrations 350 were similar to the concentrations of α-pinene and β-pinene. Accounting for about 100 times faster α-terpinene consumption

in reactions with OH and $O_3$ compared to other terpenes (Atkinson et al., 2006; IUPAC, 2020) that would imply about 100 times larger α-terpinene emission rate at the measurement site, what is unlikely considering observed compositions of monoterpene emissions of biogenic origin (Geron et al., 2000). Being well outside of the uncertainty of the $H_2SO_4$ measurements the contribution from α-terpinene was therefore excluded from consideration in this work.

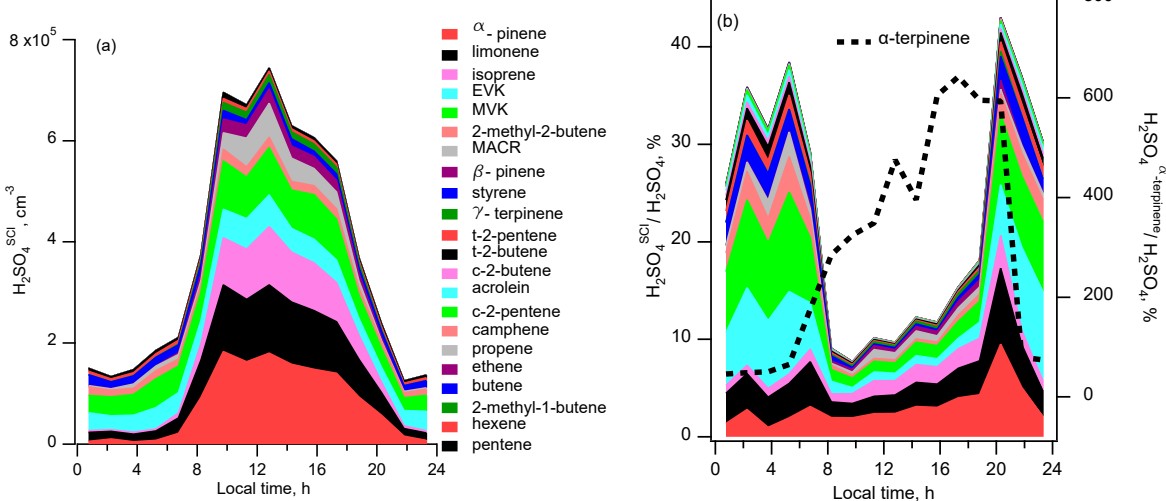

**Figure 4**. Diel profiles of the mean calculated $H_2SO_4$ concentrations produced from ozonolysis of different VOCs: (a) - filled regions correspond to $[H_2SO_4]^{SCI}$ produced by ozonolysis of different VOCs; (b) - profiles of ratios of $[H_2SO_4]^{SCI}$ from different VOCs to the measured $[H_2SO_4]$. Black dotted line in (b) corresponds to $[H_2SO_4]^{SCI}$ produced by the ozonolysis of α-terpinene

(right axis).

The calculated concentration of the sum of $H_2SO_4{}^{SCI}$ reaches a maximum of about $7\times10^5$ molecules cm$^{-3}$ around midday and goes down to about $1.5\times10^5$ molecules cm$^{-3}$ during the night. The largest estimated contribution to the $H_2SO_4{}^{SCI}$ is from α-pinene, limonene and isoprene during the day and from MVK and EVK at the night-time (Fig. 4).

The largest contribution to the calculated sum of SCIs was from the *syn* form of acetaldehyde oxide, Z-CH₃CHOO (2), E-MVK-oxide (10) from isoprene, oxo-substituted *E*-(C(O)R)CHOO (31) (with R=H, CH₃, and C₂H₅ for the SCI from acrolein, MVK and EVK, respectively), phenyl-substituted carbonyl oxide PhCHOO (35) from styrene and Z-pinonaldehyde-K-oxide (13) from α-pinene (the SCIs numbering given in parenthesis corresponds to the SCIs numbering in Tables S4 and S5). The night-time and the daytime mean concentrations of the sum of all SCIs were about $10^3$ molecule cm$^{-3}$ and $3\times10^3$ molecule cm$^{-3}$,

respectively. *Z*-pinonaldehyde-K-oxide (13) (from α-pinene) and *E*-(C(O)R)CHOO (31) (from MVK, EVK and acrolein) were also the major SCIs producing $H_2SO_4$. In addition, comparable contribution to $H_2SO_4$ formation was from *Z*-(CHR$_a$R$_b$)(CH₃)COO (21) (from limonene ozonolysis) and *E*-MVK-oxide (10) (from isoprene ozonolysis). Detailed information is given in Supplement Fig. S10.

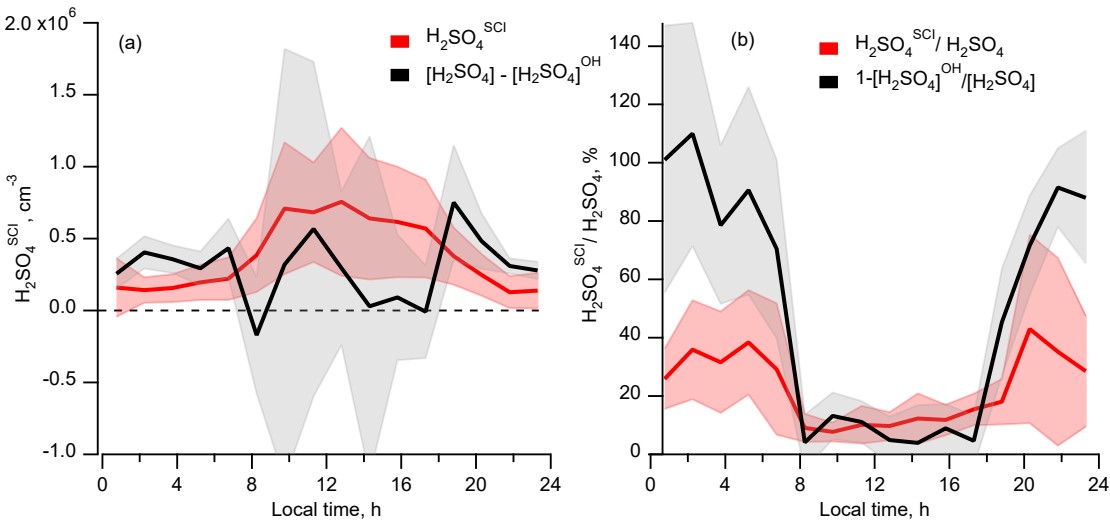

**Figure 5**. Comparison of the sulfuric acid produced by oxidation of $SO_2$ by SCIs, $H_2SO_4^{SCI}$, and missing source of $H_2SO_4$ derived from the difference of the measured $[H_2SO_4]$ and $[H_2SO_4]^{OH}$. (a) Diel profiles of the mean $[H_2SO_4]^{SCI}$ (red) and $[H_2SO_4]-[H_2SO_4]^{OH}$ (black). (b) Diel profiles of the mean of the relative contributions $[H_2SO_4]^{SCI}/[H_2SO_4]$ (red) and $1-[H_2SO_4]^{OH}/[H_2SO_4]$ (black). Shaded areas correspond to $\pm1\sigma$ standard deviation.

    Comparing the calculated $[H_2SO_4]^{SCI}$ with the missing $H_2SO_4$ source derived from the difference of the measured $[H_2SO_4]$ and calculated $[H_2SO_4]^{OH}$, Fig. 5a shows that compared to the difference of $[H_2SO_4] - [H_2SO_4]^{OH}$ representing the missing $H_2SO_4$ source, the $[H_2SO_4]^{SCI}$ is lower during the night and higher during the day with a difference of several $10^5$ molecule cm$^{-3}$ (Table 1). The large variance of the average $[H_2SO_4] - [H_2SO_4]^{OH}$ difference presented in Table 1 is due to the large

scattering of these values derived either from a small difference of large OH and $H_2SO_4$ signals or from the measurements of OH and $H_2SO_4$ concentrations close to their lower detection limits, as well as due to a stochastic day to day natural variation.

    Shown in Fig. 4b, the calculated dial profile of the relative $H_2SO_4^{SCI}$ to $H_2SO_4$ contribution resembles the dial profile of a missing $H_2SO_4$ source derived from the relative concentration difference of $H_2SO_4$ and $H_2SO_4^{OH}$, i.e. $1-[H_2SO_4]^{OH}/[H_2SO_4]$. Both profiles exhibit a shallow well from about 8:00h to 18:00h during the day with higher relative contribution during the

390 night. According to the $[H_2SO_4]^{SCI}$ calculations, the SCIs contribution is about three times higher during the night constituting $(38\pm24)\%$ compared to $(12\pm6)\%$ during daytime. The calculated $[H_2SO_4]^{SCI}$ contribution is significantly lower than the night-time missing source of $(91\pm2)\%$ and is in good agreement with the missing $(14\pm4)\%$ during the day (Table 1). The large unexplained formation of $H_2SO_4$ during the night can be related to the large uncertainty on the calculated SCI concentrations and the rates of $H_2SO_4$ production from their reactions with $SO_2$ (Vereecken et al., 2017), as well as to an unaccounted

contribution from the ozonolysis of some unsaturated compounds not measured during the campaign.



## 4 Discussion

### 4.1 Validity of $H_2SO_4$ steady state conditions

Estimated concentrations of $H_2SO_4$ produced in reactions of $SO_2$ with OH and with SCIs are based on the assumption of a steady state between production and loss pathways of $H_2SO_4$. All variations of the parameters influencing the $H_2SO_4$ concentration in the air masses incoming to the measurements site are assumed to occur on a time scale that is longer than the lifetime of $H_2SO_4$ estimated on the basis of the local aerosol measurements. These conditions are not obviously fulfilled at the Ersa site because the air masses arriving at the island may become influenced by local biogenic emissions, possibly leading to varying concentrations of these compounds along the trajectory from the coast to the measurement location (due to non-uniformly distributed emissions, development of vertical profile of VOCs and other reasons). As the production rate of the SCIs is proportional to the corresponding concentrations of the unsaturated compounds the variation of VOCs may induce variations in the SCIs concentrations. The concentration of OH could also be affected by varying VOCs concentrations considering that the lifetime of OH at the measurements site was about 0.2 s with half of the OH reactivity explained by the local biogenic emissions (Zannoni et al., 2017).

The steady state conditions were hardly disturbed by variations of the concentrations of $SO_2$ and $O_3$, species of non-local origin with the lifetime longer than the lifetime of $H_2SO_4$. Concerning the $H_2SO_4$ condensation sink, a variation in the aerosol particle number concentration and size distribution could be present. Recent studies evidenced that highly oxygenated molecules (HOMs) from the ozonolysis of monoterpenes may initiate new particle formation and its fast initial growth with the growth rates of tens of nm per hour (Stolzenburg et al., 2018; Tröstl et al., 2016). However, these variation are still slow compared to the $H_2SO_4$ lifetime of several minutes.

The time scales for the variations of VOCs, SCI, OH and *CS* may depend on the distribution of the biogenic emissions, wind speed, conditions of turbulent mixing and others. Without information about all these details we can only estimate the maximum time for such variations corresponding to the time of air masses presence over the land. To fulfill the steady state conditions this presence time has to be at least longer than the lifetime of $H_2SO_4$. The prevailing wind directions were from the NE and, predominantly, from the SW. The SW direction corresponds to the shortest presence times because of the shortest distance to the coastline and the highest wind speeds corresponding to this wind directions (Supplement Fig. S11, S12). On average, the presence time over the land, $t_{pr}$, significantly exceeded the $H_2SO_4$ lifetime, $t_{H2SO4}$, with the median value of the ratio of $t_{pr} / t_{H2SO4}$ being of 7.7 (4.3 – 17.9 interquartile range). To test an influence of the presence time on the calculated $H_2SO_4$ budget, the dependencies between the relative difference $1-[H_2SO_4]^{OH}/[H_2SO_4]$ and the relative contribution $[H_2SO_4]^{SCI}/[H_2SO_4]$ were analyzed with a filtered data using criteria $t_{pr} / t_{H2SO4} > 1$ and $t_{pr} / t_{H2SO4} > 10$. No significant difference were found compared with the unfiltered data, as it is also illustrated in Fig. 6 showing no apparent correlation between the relative contribution of the OH+$SO_2$ source and the ratio of $t_{pr} / t_{H2SO4}$. Similarly, no apparent effect was found by filtering the data according to the wind speed and the wind direction (not presented). Finally, although relatively short time scale spatial variability of OH or SCIs concentrations around the measurement site cannot be excluded, an apparent





independence on the presence time or the wind speed support the validity of the assumption used here regarding $H_2SO_4$ steady

state conditions.

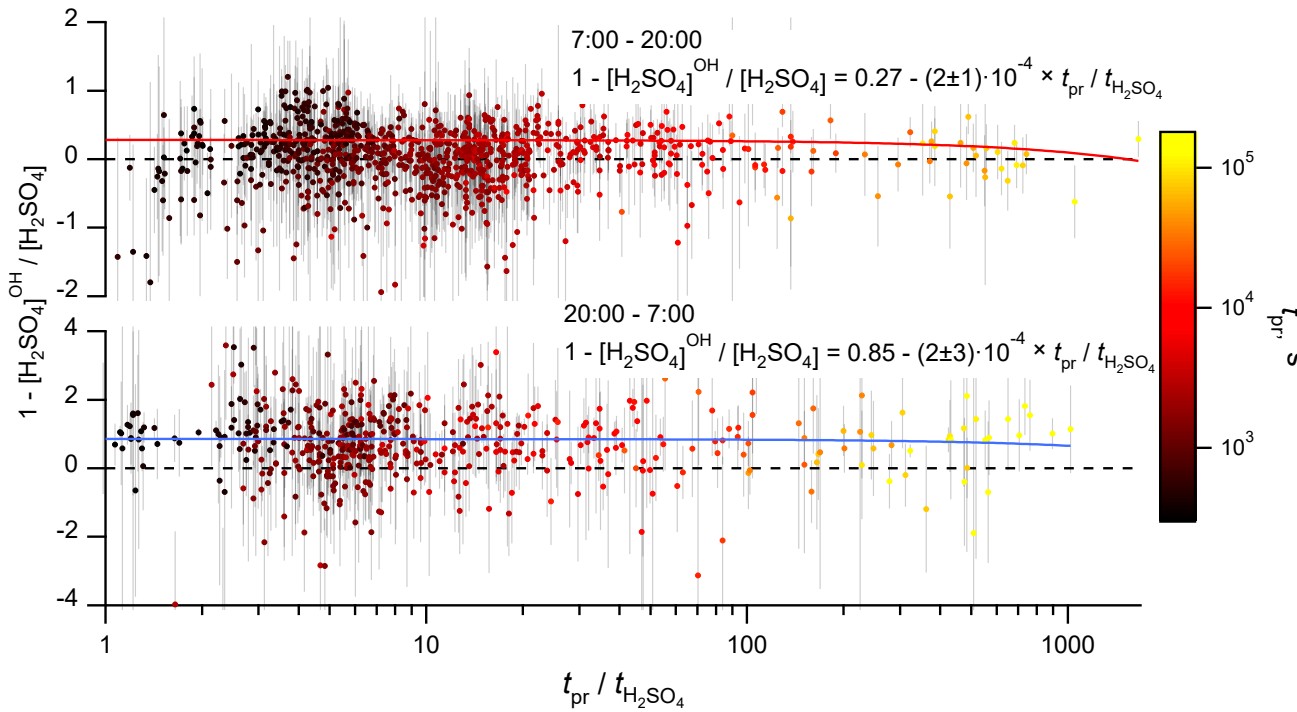

**Figure 6**. Dependence of the normalized difference of the $H_2SO_4$ concentrations observed and produced from $SO_2+OH$, 1-$[H_2SO_4]^{OH}/[H_2SO_4]$, on the ratio of presence time over the land to the lifetime of $H_2SO_4$, $t_{pr}/t_{H2SO4}$, colored by the presence time over the land, $t_{pr}$. Upper and lower figures correspond to the daytime and the night-time data, respectively. Solid lines

correspond to linear regression fits.

## 4.2 OH+SO₂ reaction rate constant

In this work the rate coefficient for the reaction of OH with $SO_2$, $k_1=8.06\times10^{-13}$ cm³ molecule⁻¹ s⁻¹ (at 760 torr and 298 K), was taken from the last published IUPAC recommendation (Atkinson et al., 2004). The two latest JPL evaluations recommend an about 20% larger rate constant of $9.6\times10^{-13}$ cm³ molecule⁻¹ s⁻¹ (Burkholder et al., 2015, 2019). Using the rate constant from

the JPL evaluations in this work would result in about 20% larger $H_2SO_4$ production rate via $OH+SO_2$. This would yield a closed $H_2SO_4$ budget with only $OH+SO_2$ source during the day and about two times lower contribution from a missing source during the night.

In a recent study of Blitz et al. (2017a, 2017b) a significantly lower rate constant of $5.8\times10^{-13}$ cm³ molecule⁻¹ s⁻¹ was derived from experiments with vibrationally excited OH (v=1,2,3)+$SO_2$ and using the master equation analysis of the pressure and temperature dependence of their own and some others experimental $OH+SO_2$ reaction rate constants. An even lower rate

constant of $4.8\times10^{-13}$ cm³ molecule⁻¹ s⁻¹ has been derived by (Medeiros et al., 2018) applying more detailed master equation



analysis of experimental data from Blitz et al. (2017a, 2017b) and some other data. These recent results have been discussed but not included in the latest JPL evaluation (Burkholder et al., 2019).

Using the lower rate constant from Medeiros et al. (2018) in our study would result in about 2 times reduced $H_2SO_4$ production by oxidation of $SO_2$ by OH and would invoke either significantly larger contribution from an additional $H_2SO_4$ source or a lower $H_2SO_4$ uptake coefficient, of about 0.5 instead of unity.

In previous field studies of the $H_2SO_4$ budget listed in Supplement Table S1, the OH+$SO_2$ production rate of $H_2SO_4$ was calculated using $k_1$ rate coefficient in the range of $(8.5 - 12) \times 10^{-13}$ $cm^3$ $molecule^{-1}$ $s^{-1}$. Reanalysis of these data using the lower $k_1$ would lead to a conclusion that the $H_2SO_4$ budget was never observed to be closed with the uptake coefficient close to unity.

On the other hand, the contribution of the $SO_2$ + SCIs source deduced in Vereecken et al. (2017) using a model with the lower $k_1$ rate constant from Blitz et al. (2017a, 2017b) would be significantly reduced employing the $k_1$ from IUPAC or JPL recommendations: from 7% to negligible in rural environment and from 70% to about 30% over tropical regions.

### 4.3 $H_2SO_4$ loss

The mass accommodation coefficient of unity used in this work was measured by (Hanson, 2005) for the $H_2SO_4$ uptake on
5 - 20 nm diameter particles composed of water and sulfuric acid. The efficient uptake of $H_2SO_4$ is supported by other studies where the accommodation coefficients of about 0.7 were determined for the uptake on liquid sulfuric acid (Pöschl et al., 1998) and on ammonium sulfate and sodium chloride particles (Jefferson et al., 1997). Lower mass accommodation coefficients in the range of 0.2-0.3 were determined for the uptake on hydrocarbon coated particles (Jefferson et al., 1997) suggesting that the uptake coefficient may depend on aerosol composition. Considering the measurements uncertainty, the results obtained in this
work are consistent with the accommodation coefficient in the range from about 0.8 to 1. At lower uptake values the OH+$SO_2$ source would significantly override the calculated $H_2SO_4$ loss during the day. At the same time, the apparently missing $H_2SO_4$ source during the night can be explained by a lower uptake coefficient, down to about 0.5.

Another possible loss mechanism of $H_2SO_4$ can be via collisions of sulfuric acid molecules leading to new particles formation (NPF) in the atmosphere. For some atmospheric conditions like in the presence of high concentrations of base
atmospheric components, e.g. ammonia and amines (Almeida et al., 2013), stabilizing the $H_2SO_4$ dimer and larger clusters, the nucleation may proceed at collisionaly limited rate corresponding to an effective bimolecular $H_2SO_4$ loss rate coefficient of about $4 \times 10^{-10}$ $cm^3$ $molecule^{-1}$ $s^{-1}$ (Kürten et al., 2014). Such conditions might have been encountered in highly polluted industrial and urban environments, regions influenced by strong agricultural emissions, and in chamber experiments (Kürten et al., 2016, 2018; Yao et al., 2018). Removal of $H_2SO_4$ with this rate constant would significantly contribute to its loss during
ChArMEx, increasing it by about a factor of 2 for the largest observed $H_2SO_4$ concentrations. During the ChArMEx/SAFMED experiment, the $H_2SO_4$ loss rate assuming sulphuric acid initiated kinetic particle formation can be estimated from several episodes of the NPF observed during the campaign (on days from 29/07 to 2/09; (Berland et al., 2017)). Fig. S13 shows example of NPF observed around midday of the 31st of July. For this day the profiles of $H_2SO_4$ and $N_{10-15}$ are similar with a time delay of about 2 hours for the maximum of the size distribution of 15 nm. This allows us to assume that the observed





NPF on this day could be related to the evolution of the $H_2SO_4$ concentration. The growth rates (GR) and 12 nm particle apparent formation rates from the SMPS measurements are 3.1 nm h$^{-1}$ and $8.2\times10^{-3}$ cm$^{-3}$ s$^{-1}$, respectively. GR was calculated using a maximum-concentration method (Kulmala et al., 2012). Using the approach of Kerminen and Kulmala (2002) and Sihto et al. (2006) to derive the rate of formation of critical cluster of size 1 nm, $J_1$, we obtain around 1 cm$^{-3}$ s$^{-1}$ at $H_2SO_4$ concentration of around $10^7$ molecule cm$^{-3}$. Assuming kinetically limited nucleation mechanism in which the critical cluster

contains two sulfuric acid molecules, this corresponds to the bimolecular rate constant of $5\times10^{-13}$ for the removal of $H_2SO_4$, which would correspond to a negligible $H_2SO_4$ loss compared to the condensation on existing particles. Similar rates, several orders of magnitude below the collision limited rate, were found in other diverse continental and marine atmospheric environments (Kuang et al., 2008). Besides, the GR estimated from the time delay between $H_2SO_4$ profile and corresponding increase in $N_{10-15}$ particles is about 2 times faster than derived from the particle size growth rate. Using this latter GR would

result in an even slower particle formation. Finally, analysis of particles origin and their chemical composition at the measurement site made with ATOMF MS indicate that even if the traces of amines were present they were of remote origin and not present on the measurement site (Arndt et al., 2017).

### 4.4 SCIs interference with OH and H₂SO₄ measurements

       High concentrations of $SO_2$ used in the chemical conversion reactor of CIMS instruments for the conversion of OH into

$H_2SO_4$ may lead to an interference with OH and $H_2SO_4$ measurements due to an artificial generation of $H_2SO_4$ in reactions of SCIs with $SO_2$ inside the reactor. When the reactant used as a scavenger of OH does not react fast with SCIs, e.g. using propane, the contribution of the artificially formed $H_2SO_4$ is the same for the OH measurements in the background (BG) and the OH signal + background modes (OH+BG), if the $SO_2$ injection position rests unchanged for the measurements in both modes. In this case the OH signal derived from the difference of the OH+BG and the BG signals is free from the artificial $H_2SO_4$

formation. However, the presence of an additional $SO_2$ oxidant not efficiently removed by OH scavenger may lead to a significant increase of the BG level as it was observed in previous field measurements (Berresheim et al., 2014; Mauldin III et al., 2012).

       In the instrument used in this study the OH is scavenged with $NO_2$. The rate constant for the reaction of $NO_2$ with $CH_2OO$, $CH_3CHOO$ and $(CH_3)_2COO$ stabilized CIs was found to be of about $2 \times 10^{-12}$ cm$^3$ molecule$^{-1}$ s$^{-1}$ (Chhantyal-Pun et al., 2017;

Stone et al., 2014; Taatjes et al., 2013), which is about 10 – 100 times smaller than typical rate constants of the reactions of SCIs with $SO_2$ (Supplement Table S5). As the ratio of $[NO_2]/[^{34}SO_2]$ in the reactor was about 100, the SCIs in the reactor may react with similar rates with both $SO_2$ and $NO_2$ contributing to the BG signal. In addition, as the $NO_2$ and $SO_2$ injection positions are different for the BG and OH+BG modes (see Supplement Fig. S1 for details), the OH signal derived from the OH+BG and the BG difference may also be influenced by the artificial $H_2^{34}SO_4$ production in reactions of SCIs with $SO_2$.

As presented in Fig. 7, the BG signal observed during ChArMEx showed a diel profile similar to that of VOCs or OH (Fig. 1 and 2) with maximum at noon and minimum during the night. The peak noon and night-time BG signals reached levels corresponding to OH concentrations of about $5\times10^6$ molecule cm$^{-3}$ and $1\times10^6$ molecule cm$^{-3}$, respectively. These BG levels



are significantly higher than values of about $(1 - 5) \times 10^5$ molecule cm$^{-3}$ typically observed with the present instrument during

calibration with Zero air or during measurements in clean air with low VOCs concentrations, e.g. in Antarctica (Kukui et al.,

2014).

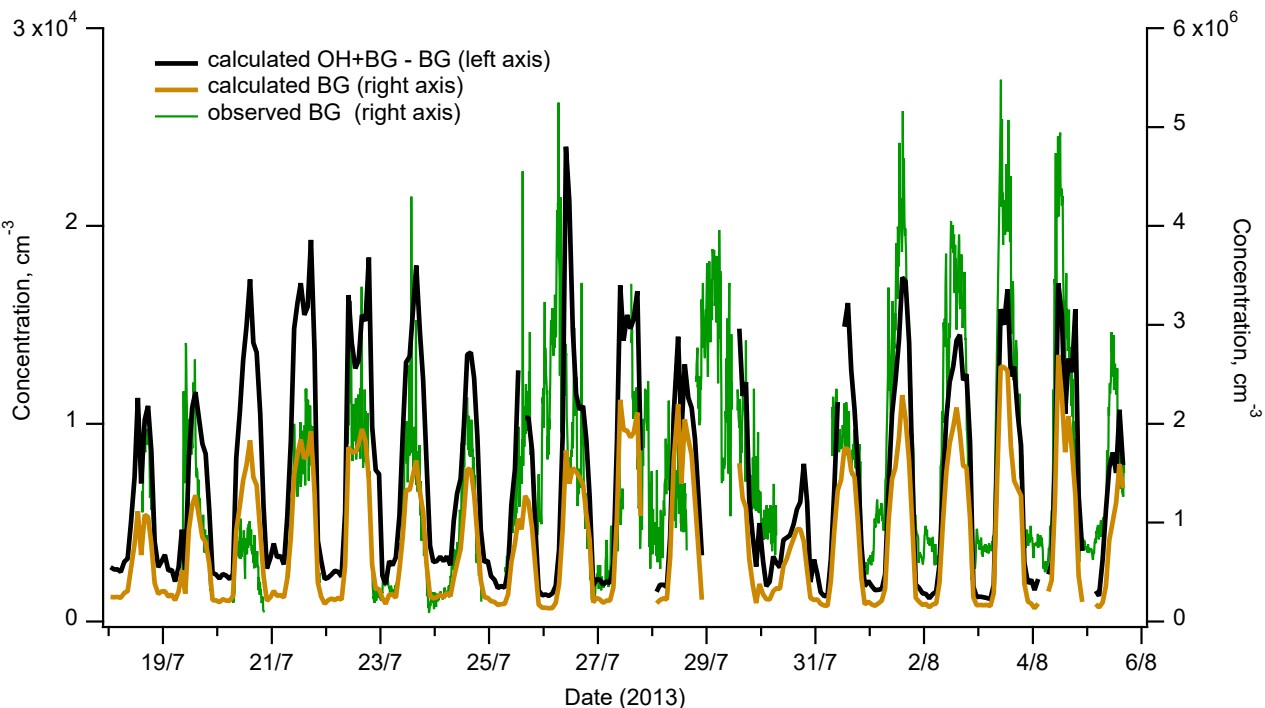

**Figure 7.** Interference with OH measurements from the H$_2$SO$_4$ produced in the reactor by reactions of SO$_2$ with SCIs:

calculated contribution to measured OH (red line, left axis) and comparison of the observed background (BG) (blue line, right

axis) and the BG calculated assuming its origin from the reactions of SCIs with SO$_2$ inside the reactor (black line, right axis).

To examine if the observed high BG levels can be explained by the SO$_2$ reaction with SCIs the H$_2$SO$_4$ concentration

produced in the reactor in reactions of the sum of SCIs with SO$_2$, [H$_2$$^{34}$SO$_4$]$^{SCI/R}$, was calculated for BG and OH+BG modes

using Eq. (4), which is similar to Eq. (2) but neglects the H$_2$SO$_4$ losses in the reactor:

$$\left[ H_2{}^{34}SO_4 \right]^{SCI/R} = \sum_{R_j} \left\{ \sum_i k_2^i \cdot \left[ SCI^i \right]^{R_j} \right\} \times \left[ {}^{34}SO_2 \right]^{R_j} \times t_{R_j} ,
\tag{4}$$

where the index $R_j$ corresponds to different parts of the reactor, $\left[ SCI^i \right]^{R_j}$ and $\left[ {}^{34}SO_2 \right]^{R_j}$ are the concentrations and $t_{R_j}$ is the

bulk flow time in the corresponding parts of the reactor (see also Supplement Fig. S1). The concentrations of SCIs inside the

reactor generated by ozonolysis of the measured unsaturated VOCs were calculated similarly to Eq. (3) but with accounting



for the additional SCIs loss in reactions with $SO_2$ and $NO_2$ injected into the reactor. The concentrations of VOCs, $O_3$ and $H_2O$ inside the reactor were calculated using the ambient measurements with accounting for a dilution inside the reactor.

The concentration of $O_3$ in the IMR was corrected by accounting for ozone generated in the corona discharge ion source and added to the IMR with the flow of primary ions. Amount of $O_3$ generated by the ion source depends on the ion source operating configuration, i.e., flow rates of injected mixtures, composition of the mixtures and potentials of the ion source electrodes. The concentration of $O_3$ in the IMR was not monitored during the campaign, but according to later checks this concentration was of $2 \pm 1$ ppm and this value was used for the estimation of $[H_2{}^{34}SO_4]^{SCI/R}$.

As shown in Fig. 7, the calculated and the observed BGs exhibit similar variability and correspond to comparable concentration levels allowing us to suggest that the observed elevated BG levels were related to the $SO_2$ oxidation by SCIs in the reactor. This hypothesis is supported by results of later experiments on the ozonolysis of terpenes conducted in an environmental chamber where a strong dependence of the observed BG level on the SCIs production rate was confirmed, as shown in Supplement Fig. S14 for the ozonolysis of α-pinene.

Concerning a possible interference of the $SO_2$+SCIs reaction with the OH measurements, Fig. 7 shows that during the campaign, this interference was negligible because the difference of $[H_2{}^{34}SO_4]^{SCI/R}$ produced in the OH+BG and the BG modes corresponded to OH concentration lower than $2\times10^4$ molecule cm$^{-3}$, about 10 times lower than the OH lower detection limit. Notably, this contribution is independent on the $[O_3]$ in the IMR influencing equally the OH+BG and BG measurements.

## 5 Conclusions

The formation of $H_2SO_4$ was observed at the Ersa station in northern Corsica, a site influenced by local emissions of biogenic VOCs. The $H_2SO_4$ concentration reached $10^7$ molecule cm$^{-3}$ at midday and was around $5\times10^5$ molecule cm$^{-3}$ during the night. Based on the OH, $H_2SO_4$, $SO_2$ and particle number density measurements and assuming validity of a steady state between $H_2SO_4$ production and its loss by condensation on existing aerosol particles with a unity accommodation coefficient, we have found that the contribution of the $SO_2$ + OH reaction accounts for $(86 \pm 4)$ % and only for $(9 \pm 2)$ % of the $H_2SO_4$ production during the day and night, respectively. The given accuracy of these values has been estimated without accounting for the large uncertainty in the OH + $SO_2$ reaction rate coefficient, which results in a larger uncertainty in the derived here contribution to $H_2SO_4$ formation from the OH+$SO_2$ source, from about a factor of 1.5 its overestimation to about 20% its underestimation.

Estimating the $H_2SO_4$ production from the $SO_2$ oxidation by SCIs, we conclude that despite the low calculated SCIs concentrations ($(1-3) \times 10^3$ molecule cm$^{-3}$ for the sum of SCIs), this source may explain about 10% of the $H_2SO_4$ formation during the day and represents a major source of $H_2SO_4$ accounting for about 40% of its formation during the night. The sum of the $H_2SO_4$ production rates via $SO_2$+OH and $SO_2$+SCIs correspond to a closure of the $H_2SO_4$ budget during the day, but seem to underestimate by 50% the $H_2SO_4$ production during the night, with the latter being possibly related to uncertainties in





the used in this work kinetic parameters, an unaccounted contribution from the ozonolysis of some unsaturated compounds not measured during the campaign, as well as to some yet unidentified $H_2SO_4$ production mechanisms during night-time.

Both the daytime and night-time results of this study indicate that the oxidation of $SO_2$ by SCIs may be an important source of $H_2SO_4$ in VOCs-rich environments, especially during night-time.

**Author's contributions**

AK designed the CIMS instrument, performed the OH and $H_2SO_4$ measurements, analyzed the data and wrote the paper. MC participated in the design of the new calibration system. All other coauthors participated in data collection. All coauthors participated in paper discussion.

**Competing interests**

The authors declare that they have no conflict of interest

**Data availability**

Data is available upon request from the authors.

**Acknowledgements**

The authors would like to thank Thierry Vincent, Stephane Chevrier and Gilles Chalumeau from LPC2E for logistical help in preparation and during the campaign, Karine Sartelet from CEREA for managing the SAFMED project, Jean Sciare from
LSCE (now at CyI) for managing the campaign site and providing meteorology data, Eric Hamonou and François Dulac from LSCE for organizing the ChArMEx campaign and for managing and coordinating the ChArMEx project, François Dulac for editorial corrections and suggestions.

**Financial support**

This research has received funding from the French National Research Agency (ANR) projects SAFMED (grant ANR-12-
BS06-0013). This work is part of the ChArMEx project supported by ADEME, CEA, CNRS-INSU and Météo-France through the multidisciplinary program MISTRALS (Mediterranean Integrated Studies aT Regional And Local Scales). The station at Ersa was partly supported by the CORSiCA project funded by the Collectivité Territoriale de Corse (CTC) through the Fonds Européen de Développement Régional of the European Operational Program 2007–2013 and the Contrat de Plan Etat-Région. This project was also supported by the CaPPA project (Chemical and Physical Properties of the Atmosphere), funded by the





French National Research Agency (ANR) through the PIA (Programme d'Investissement d'Avenir) under contract ANR-11-LABX-0005-01 and by the Regional Council Nord-Pas de Calais and the European Funds for Regional Economic Development (FEDER). This work was also supported by the LABEX VOLTAIRE ANR-10-LABX-100-01 (2011–20) and the PIVOTS project provided by the Région Centre—Val de Loire (ARD 2020 program and CPER 2015–2020).

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
