# Peer review of "Role of Criegee intermediates in the formation of sulfuric acid at a Mediterranean (Cape Corsica) site under influence of biogenic emissions"

_Atmospheric Chemistry and Physics, 2021_

## Referee Comment (RC2)

The authors describe results from a field measurement campaign held in summer 2013 at Cape Corsica. The intention is to figure out the relative importance of the different $H_2SO_4$ production channels, i.e. either via OH + $SO_2$ or CI(Criegee Intermediate) + $SO_2$. $H_2SO_4$ and OH radicals have been directly measured by means of a nitrate-CIMS. Overall steady-state CI concentrations were estimated based on measured alkene concentrations considering the unimolecular CI loss as well as bimolecular CI reactions with water vapour and the water dimer. All needed rate coefficients were taken from the recent literature. For comparison, $H_2SO_4$ produced via the OH + $SO_2$ channel has been calculated using k(OH+$SO_2$) from the IUPAC recommendation from 2004 and $H_2SO_4$ from CI + $SO_2$ in an analogous way using actual rate coefficients.

The authors came to the conclusion that at daytime 86±4% of the observed $H_2SO_4$ are formed via the OH + $SO_2$ channel, and only 9±2% during night. The corresponding data from the CI + $SO_2$ channel are 12±6% and 38±24%, respectively. Thus, at least at daytime the observed $H_2SO_4$ is well explained by both reaction channels (although I think that the range of uncertainties must be clearly bigger).

All in all, it is a very nice work, easy to understand and well structured. This manuscript is suitable for publication in ACP. Some minor points should be considered before final acceptance:

- The authors consider OH + $SO_2$ and CI + $SO_2$ for $SO_3$ production, and subsequent $H_2SO_4$ formation, only. What about the possible direct route via oxidation of reduced-sulfur compounds (DMS etc.) as discussed by Berresheim et al., 2014, 10.5194/acp-14-12209-2014, and in a couple of other papers? Is a contribution of this direct route totally negligible at this coastal site? Please comment.
- I´m struggling a bit with the used k(OH+$SO_2$) from the IUPAC 2004 recommendation. The more recent value by Blitz et al., 2017, 10.1021/acs.jpca.7b01295, is clearly smaller. Consequently, also the $H_2SO_4$ production from this channel will become smaller worsening the good agreement between measurement and calculation at daytime. But, a better agreement with any calculations is not an argument for a special parameter. So, it would be fine, if the authors could discuss the results based on both k(OH+$SO_2$)´s more in detail, not only very briefly as in paragraph 4.2.
- Line 80: "monomolecular"? unimolecular

---

## Author Response (AR1)

We would like to thank reviewers for their reviews of this manuscript and providing helpful comments and suggestions. Here we provide responses to these referee comments: referee comments are shown in black, and our responses in blue. The parts of the manuscript revised in response to comments are in blue italics.

**Anonymous Referee #2**

This study focuses on the investigation of the $H_2SO_4$ budget in a remote coastal environment by using measured $H_2SO_4$ and OH radical with a CIMS instrument. The study investigates whether SCI have an impact in the $H_2SO_4$ formation for this environment and finds that they contribute to less than 10% to the $H_2SO_4$ during day and up to 40% during the night. 50% of the $H_2SO_4$ observed during the night cannot be explained.

The paper is well written and well structured and makes use of the most up to date literature values for the SCI chemistry to assess their impact on the $H_2SO_4$ formation. It extensively discusses the relatively large uncertainties that pertains to both the SCI, the loss rate for $H_2SO_4$ as well as the uncertainties on the rate coefficient between OH and $SO_2$. The latter I find rather interesting as the discrepancies with available (and recommended) rates are large and introduce a big bias when trying to investigate additional sources for the $H_2SO_4$.

I recommend publication after the following comments are considered.

Specific comments:

Check the formatting of the citations. Often it is formatted wrongly when in the text (especially in the SI).

Citation formatting is corrected on the lines 79, 83, 249, 446, 459 of the manuscript and on the pages 2, 4, 8, 14, 16.

Please add the page number to the SI.

We have added the page number in all the references to the SI.

Page 2, Line 45: I do not understand the sentence "…a noticeable fraction of nucleation mode particle's growth…"

Rephrased as:

"*$H_2SO_4$ is considered as a major precursor of newly formed atmospheric nucleation-mode particles and may play a significant role in their subsequent growth (Boy et al., 2005; Smith et al., 2005; Zhang et al., 2012).*

We have also added here a reference to:

*Zhang, R., Khalizov, A., Wang, L., Hu, M., and Xu, W.: Nucleation and Growth of Nanoparticles in the Atmosphere, Chem. Rev., 112, 1957-2011, doi:10.1021/cr2001756, 2012.*

Page 3, Line70: together with the formation of a SCI there is always the formation of a carbonyl compound.

We have added this remark as follows:

"*Subsequent rapid cleavage of either of the O-O bonds leads to the formation of a carbonyl compound and chemically activated CIs, …*"

Page 4, Lines 102-103: There is experimental evidence that the formation of $SO_3$ from the reaction of SCI from β-pinene and $SO_2$ is pretty much immediate (Ahrens et al., 2014) so that I would think there is not much doubt about it also for larger SCI.

We have added this information as follows:

"*For larger SCIs with expected longer SOZ lifetime the $SO_3$ yield may depend on the SOZ fate in the atmosphere with respect to its decomposition or further reactions, e.g. with $H_2O$, (Kuwata et al., 2015; Vereecken et al., 2012), although*

*a large yield of sulfur trioxide exceeding 80% was observed by Ahrens et al. (2014) for the large SCI formed during the ozonolysis of β-pinene."*

Page 4, Line 110: I would also add the reference to the paper by Novelli et al. (2017) which came up with a similar estimate from measured data.

*We have added the reference to:*

*Novelli, A., Hens, K., Ernest, C. T., Martinez, M., Nölscher, A. C., Sinha, V., Paasonen, P., Petäjä, T., Sipilä, M., Elste, T., Plass-Dülmer, C., Phillips, G. J., Kubistin, D., Williams, J., Vereecken, L., Lelieveld, J. and Harder, H.: Estimating the atmospheric concentration of Criegee intermediates and their possible interference in a FAGE-LIF instrument, Atmos. Chem. Phys., 17(12), 7807–7826, doi:10.5194/acp-17-7807-2017, 2017.*

Section 2.2.1. Please be careful in the definition of ROx. Is it $OH+HO_2+RO_2$ or is the OH contribution removed? Later on (page 6 line 176 and 181), I assume, ROx becomes $RO_2$ but that is a big difference. Was the $HO_2$ contribution removed from the ROx signal or this is a typo? If the CIMS can separate between $HO_2$ and $RO_2$ it would be interesting to shortly clarify this.

*With our CIMS instrument we measure, in addition to $H_2SO_4$, either concentration of OH radical or the sum of concentrations of OH, $HO_2$ and $RO_2$ (= $RO_x$) by converting peroxy radicals ($HO_2+RO_2$) to OH in presence of added NO. As the atmospheric concentrations of OH are typically much lower compared to $HO_2+RO_2$, the second mode with measurements of $[RO_x]$ is close to measurements of peroxy radicals $[HO_2]+[RO_2]$. We do not discuss the $HO_2+RO_2$ measurements as they are not used in this work. We cannot reliably separate between $HO_2$ and $RO_2$ measurements.*

*We are thankful for your remark about using "$RO_2$" on lines 176 and 181. We have corrected the text by replacing "$RO_2$ radical" on lines 176 and 181 by "peroxy radicals", already defined as $HO_2+RO_2$ on line 158.*

Page 9, Line 275: Isn't the value of OH measured at night lower than the stated detection limit of $5x10^5$ cm$^{-3}$? How meaningful is the nightime analysis then?

*OH data are derived from 7 min integration time points resulting from averaging of the difference of about 100 individual points of background OH and signal OH measurements. From these 100 individual 1s integration points we obtain 7 min means and corresponding variances. All other OH concentrations used in the article, like 90 min averages and mean or median night time values are derived by weighted averaging of these 7 min data with calculation of corresponding data precision. The obtained uncertainty of the resulting averaged values takes into account the uncertainties and weight of individual points regardless of their values lying below or above the estimated detection limit. The individual points below LOD with large signal to noise ratio contribute to larger noise of the resulted data. These uncertainties are presented in the text and in Table 1.*

*We are aware about discussion of different approaches for statistical treatment of <LOD data, like different methods of substitution, nonparametric methods and others (Antweiler, R. C., Environ. Sci. Technol., 49, 22, 13439–13446, 2015). It appears, however, that using instrument derived <LOD data, compares well with other techniques, especially when uncertainty of the raw instrumental data can be estimated.*

*Finally, we would like to mention, that the LOD for OH was arising from the elevated background during ChArMEx campaign. For the day time it was estimated to be of $5×10^5$ molecule cm$^{-3}$. The nighttime background was about 3-4 times lower (see Figure 7), resulting in lower nighttime LOD of about $(1 − 2) ×10^5$ molecule cm$^{-3}$. We have added this correction to the text of the article (on Line 204, 205):*

*"Accordingly, the lower limits of detection at signal-to-noise ratio of 2 and a 15 min integration time were $2×10^5$ molecule cm$^{-3}$ for $H_2SO_4$ and $5×10^5$ molecule cm$^{-3}$ and $2×10^5$ molecule cm$^{-3}$ for OH daytime and nighttime measurements, respectively."*

Page 12, Line 318: It is stated that the reaction between OH and $SO_2$ under estimate the $H_2SO_4$ concentration at night. Although this is clearly visible in figure 2a, I am not so convinced it is clear from figure 1 where I have the feeling that overall the $H_2SO_4$ is well explained by the OH radical as in the night often the $H_2SO_4$ calculated from OH is missing

or has some sharp low values which might bias the median profile shown in figure 2a. What is the cause for the sharp low values for the $H_2SO_4$ calculated from OH+$SO_2$ in figure 1?

We agree that the difference of measured and calculated $H_2SO_4$ concentrations is not clearly visible in Figure 1. The sharp low values on the log-plot appear because the values of about zero are not seen on this plot. To make it clearer for the night-time data we have added an additional plot to Figure 1 for the $H_2SO_4^{mes}$ and $H_2SO_4^{OH}$ with appropriate y-axis range and using a linear scale.

The missing calculated $H_2SO_4$ data are explained by absence of night-time measurements before 23 of July, absence of $SO_2$ measurements on night 28/07 and filtering out of some data obtained under conditions of very strong fog on 28-30 of July mentioned on page 9.

The Figure 1 and the figure caption is modified as follows:

[Figure]

*“Figure 1. Time series of the observations during the ChArMEx summer 2013 campaign: OH radicals and $O_3$ (a); sulfuric acid observed, $H_2SO_4^{mes}$, and calculated assuming only $SO_2$+OH source, $H_2SO_4^{OH}$ (see Sect. 2.3) (b); $SO_2$ and condensation sink CS (c); total monoterpenes and isoprene (left axis), EVK (right axis) (d). For clarity, comparison of the measured $H_2SO_4^{mes}$ and calculated $H_2SO_4^{OH}$ concentrations is presented in (b) using two plots, with logarithmic and linear scales on the y-axis.”*

Page 14, Line 352: which instead of what

Done

Page 15, Figure 5a: Shouldn't the difference between the measured $H_2SO_4$ and the $H_2SO_4$ calculated from the contribution of the OH radical always be higher or equal to the $H_2SO_4$ calculated from the SCI? Also, it looks relatively stable over the whole day…more as if affected by some scaling factor than additional chemistry. Once considering all the different uncertainties I am not so sure that I would conclude the abstract stating that SCI are an important source of $H_2SO_4$ in SCI rich environments. Also, as the reaction with acids (which is fast) is not included as a loss rate for the SCI, what described here is an upper limit.

We agree that actual contribution of SCI to $H_2SO_4$ is lower than (or equal to) the difference of the measured $H_2SO_4^{mes}$ and calculated $H_2SO_4^{OH}$. However, accounting for the large uncertainty of the calculated $H_2SO_4^{SCI}$ produced by SCI, estimated to be of an order of magnitude as mentioned on page 9, the overestimation of the $H_2SO_4^{SCI}$ compared to $(H_2SO_4^{mes} - H_2SO_4^{OH})$ during the day shown in Figure 5 seems to be acceptable. Please note also, that $H_2SO_4^{SCI}$ and $(H_2SO_4^{mes} - H_2SO_4^{OH})$ during the day are in agreement within the estimated uncertainty of the measurements.

We agree that neglecting the loss of SCI in reaction with acids results in overestimation of $H_2SO_4^{SCI}$ (typically, less than 10% estimated by Vereecken et al. (2017)). At the same time, the calculated $H_2SO_4^{SCI}$ was probably underestimated because of incomplete set of SCIs taken into account and, thus, in our opinion, cannot be considered as an upper limit.

Our conclusion about possible importance of the SCI source of $H_2SO_4$ is based mostly on the observed difference of $(H_2SO_4^{mes} - H_2SO_4^{OH})$, which can be at least partly explained with the estimated $H_2SO_4^{SCI}$, within the large estimated uncertainties.

Page 18, Lines 455-457: I am not sure I follow here. If the lower rate coefficient for OH + $SO_2$ as proposed by (Blitz et al., 2017a, 2017b), shouldn't the contribution of SCI to the formation of $H_2SO_4$ increase substantially?

For clarity, we have rephrased the sentence as follows:

"On the other hand, the employing the lower $k_1$ from Blitz et al. (2017a, 2017b) in model studies results in a significantly larger relative contribution of SCI to the $H_2SO_4$ formation. For example, the SCI contribution of 7% in a rural environment and of 70% in tropical regions, which were estimated assuming the lower $k_1$ by Vereecken et al. (2017), would be reduced respectively to negligible and to about 30%, if the larger $k_1$ from IUPAC or JPL was used."

**Anonymous Referee #3**

The authors describe results from a field measurement campaign held in summer 2013 at Cape Corsica. The intention is to figure out the relative importance of the different $H_2SO_4$ production channels, i.e. either via OH + $SO_2$ or CI(Criegee Intermediate) + $SO_2$. $H_2SO_4$ and OH radicals have been directly measured by means of a nitrate-CIMS. Overall steady-state CI concentrations were estimated based on measured alkene concentrations considering the unimolecular CI loss as well as bimolecular CI reactions with water vapour and the water dimer. All needed rate coefficients were taken from the recent literature. For comparison, $H_2SO_4$ produced via the OH + $SO_2$ channel has been calculated using k(OH+$SO_2$) from the IUPAC recommendation from 2004 and $H_2SO_4$ from CI + $SO_2$ in an analogous way using actual rate coefficients.

The authors came to the conclusion that at daytime 86±4% of the observed $H_2SO_4$ are formed via the OH + $SO_2$ channel, and only 9±2% during night. The corresponding data from the CI + $SO_2$ channel are 12±6% and 38±24%, respectively. Thus, at least at daytime the observed $H_2SO_4$ is well explained by both reaction channels (although I think that the range of uncertainties must be clearly bigger).

All in all, it is a very nice work, easy to understand and well structured. This manuscript is suitable for publication in ACP. Some minor points should be considered before final acceptance:

The authors consider OH + SO$_2$ and CI + SO$_2$ for SO$_3$ production, and subsequent H$_2$SO$_4$ formation, only. What about the possible direct route via oxidation of reduced-sulfur compounds (DMS etc.) as discussed by Berresheim et al., 2014, 10.5194/acp-14-12209-2014, and in a couple of other papers? Is a contribution of this direct route totally negligible at this coastal site? Please comment.

We have mentioned in the Introduction section that the oxidation of DMS or DMDS in remote coastal environments proceeding with SO$_3$ formation was suggested previously as a possible mechanism of the H$_2$SO$_4$ production (Berresheim et al., 2002, 2014; Jefferson et al., 1998). While it is possible that this route is to some extent responsible for the missing H$_2$SO$_4$ source during the night, we cannot discuss its importance in absence of DMS and other related measurements during the campaign. We can only mention here, that concentration of methane sulfonic acid (MSA), occasionally measured with our instrument during the campaign, was found to be lower than detection limit of about $10^5$ cm$^{-3}$. Considering that MSA is known as one of the products of DMS oxidation, the concentration of DMS at the measurement site was probably also very low. Furthermore, DMS concentrations was below the detection limit of PTR-MS.

I´m struggling a bit with the used k(OH+SO$_2$) from the IUPAC 2004 recommendation. The more recent value by Blitz et al., 2017, 10.1021/acs.jpca.7b01295, is clearly smaller. Consequently, also the H$_2$SO$_4$ production from this channel will become smaller worsening the good agreement between measurement and calculation at daytime. But, a better agreement with any calculations is not an argument for a special parameter. So, it would be fine, if the authors could discuss the results based on both k(OH+SO$_2$)´s more in detail, not only very briefly as in paragraph 4.2.

Our choice of the $k$(OH+SO$_2$) reaction coefficient is based on the latest IUPAC and JPL recommendations, as it is explained at the beginning of Section 4.2. We also discuss in quantitative terms what would be the impact on our conclusions of using recently suggested lower rate coefficient. This impact is straightforward, as the yield of H$_2$SO$_4$ from the reaction of OH+SO$_2$ is linearly proportional to the reaction coefficient $k$(OH+SO$_2$). We find that comparison of calculations using different rate constants would significantly overload the article without adding new information.

Line 80: "monomolecular"? unimolecular

Changed to *"unimolecular"*

**Anonymous Referee #4**

Kukui et al. investigate the potential contribution of Criegee intermediates to sulfuric acid formation in the western Mediterranean, via analysis of field measurements from the ChArMEx project. Two chemical pathways to sulfuric acid are considered: reaction of OH with SO$_2$ (+ H$_2$O + O$_2$), and the reactions of stabilized (thermalized) Criegee intermediates (SCIs) with SO$_2$. Using their measurements of OH and unsaturated VOCs, Kukui et al. estimate the contribution of each channel to measured sulfuric acid and determine that reaction of SCIs with SO$_2$ contribute ~10% of daytime and ~40% of nighttime sulfuric acid. While this analysis closes the daytime budget for sulfuric acid, ~50% of the nighttime sulfuric acid concentration remains unaccounted for. The authors conclude that SCI chemistry may be an important nighttime sulfuric acid source for VOC-rich environments.

This is a timely and interesting study that is suitable for publication in ACP. Some comments, questions, and suggestions follow.

The reaction of MACR-oxide with SO$_2$ has also recently been measured (Lin et al., Chemistry Communications, 2021).

Reference is added to the main text and the Supplement Table S5.

To what extent do you anticipate that an inability to quantify all unsaturated VOCs with your field instrument could be contributing to the significant unexplained nighttime sulfuric acid? One way to explore this could be through comparison of calculated vs. measured OH reactivity. For example, if you compare measured [VOC] to measured OH reactivity, do you have significant underprediction of OH reactivity that could be rationalized by higher [unsaturated VOC] than you were able to quantify in the field?

*The OH reactivity was measured during the ChArMEx campaign and we have added the following comment on page 15, line 395:*

*"... as well as to an unaccounted contribution from the ozonolysis of some unsaturated compounds not measured during the campaign. The latter explanation is supported by a significant daytime and night-time missing OH reactivity of about 50%, observed by Zannoni et al. (2017) during ChArMEx using the same VOCs data as in the present work. The main unaccounted species were suggested to be reactive biogenic VOCs including sesquiterpenes, oxygenated terpenes and their oxidation products. Ozonolysis of these compounds could be an additional unaccounted source of SCI in the present work."*

What parameters from the literature do you use to calculate the concentration of the water dimer vs. monomer?

*We have added corresponding reference and the following remark on Line 265 and:*

*"Concentration of $(H_2O)_2$ was calculated using an equilibrium constant for water dimer formation and its temperature dependence from Ruscic (2013)."*

*Ruscic, B.: Active Thermochemical Tables: Water and Water Dimer, J. Phys. Chem. A, 117, 11940−11953, doi:10.1021/jp403197t, 2013*

The layout of Figure 3 (positioning of inset figures) makes this figure confusing to look at.

*We have slightly modified Figure 3 by making better visible the nigh-time data on the central plot. We have also added a comment to the figure capture:*

[Figure]

*"**Figure 3**. Comparison of the measured $H_2SO_4$ and the $H_2SO_4^{OH}$ calculated accounting only for the OH+SO_2 source during the day (red) and during the night (blue). Solid lines correspond to linear regression fitting accounting for both X and Y measurement uncertainties. Dashed lines represent 1:1 ratios. Inserts, log-plot (upper) and linear plot (lower right), are added for clearer presentation of the night-time data."*

In section 4.2., the phrasing where you describe the work of Vereecken et al. (that uses the Blitz et al. rate constants) is somewhat confusing – please rephrase for clarity.

Rephrased as:

*"On the other hand, the employing the lower $k_1$ from Blitz et al. (2017a, 2017b) in model studies results in a significantly larger relative contribution of SCI to the $H_2SO_4$ formation. For example, the SCI contribution of 7% in a rural environment and of 70% in tropical regions, which were estimated assuming the lower $k_1$ by Vereecken et al. (2017), would be reduced respectively to negligible and to about 30%, if the larger $k_1$ from IUPAC or JPL was used."*

I would also suggest stating somewhere that there are, inevitably, large uncertainties in the rate coefficients for the reaction of SO2 with more complex SCI that have not yet been directly measured or calculated.

We address this question on lines 92-115 of the Introduction.  In particular, on line 112 we note that: "The uncertainty associated to the predicted SCI concentrations was estimated to be one order of magnitude (Vereecken et al., 2017), due to poorly defined SCI formation and loss rates. Even a higher uncertainty may be expected for the estimated contribution of SCIs to $SO_2$ oxidation considering not well defined reaction rate coefficients for the reaction of different SCIs with $SO_2$."

In Table S5, it is noted that the experimental results of Caravan et al. suggest faster unimolecular decomposition of Z-MVK-oxide in better agreement with the results of Vereecken et al. This is not correct – Caravan et al. actually suggest that faster decay rate in their experiments could be due to internal excitation of anti-MVK-oxide and/or a low yield of stabilized anti-MVK-oxide – both of which are experimental factors.

We agree that results of Caravan et al. do not unambiguously indicate faster decay of thermalized Z-MVK-oxide and that they may be related to some experimental factors. This question is not essential for this work and we remove the corresponding note in Table S5.

---

## Editor Decision (ED1)

Dear Dr Kukui et al.,

Thank you for addressing most of the reviewers' comments. I believe your paper provides useful and novel information on H2SO4 budget in a remote coastal environment and can be published after addressing the reviewer's #3 question.

Although I understand you have some concerns regarding '*overloading the article without adding new information*' when responding to the comment below to the reviewer #3, I feel the reviewer raised a very important point here. I suggest to address this point either by expanding your calculations using both k(OH+SO2), as stated below, or expand your discussion section, supporting the rationale for your choice, in the text, so it is clear to your article reader.

Comment from Reviewer #3 (and authors' response) :

I´m struggling a bit with the used k(OH+SO$_2$) from the IUPAC 2004 recommendation. The more recent value by Blitz et al., 2017, 10.1021/acs.jpca.7b01295, is clearly smaller. Consequently, also the H$_2$SO$_4$ production from this channel will become smaller worsening the good agreement between measurement and calculation at daytime. But, a better agreement with any calculations is not an argument for a special parameter. So, it would be fine, if the authors could discuss the results based on both k(OH+SO$_2$)'s more in detail, not only very briefly as in paragraph 4.2.

Our choice of the $k$(OH+SO$_2$) reaction coefficient is based on the latest IUPAC and JPL recommendations, as it is explained at the beginning of Section 4.2. We also discuss in quantitative terms what would be the impact on our conclusions of using recently suggested lower rate coefficient. This impact is straightforward, as the yield of H$_2$SO$_4$ from the reaction of OH+SO$_2$ is linearly proportional to the reaction coefficient $k$(OH+SO$_2$). We find that comparison of calculations using different rate constants would significantly overload the article without adding new information.

Kind regards,

Ivan Kourtchev

---

## Author Response (AR2)

Dear Editor,

Thank you for your recommendation to respond more clearly on the reviewer's #3 comment about used in our work rate coefficient of the reaction of $SO_2$ with OH. Following this recommendation, we extended the Table 1 with the results showing the contribution of the OH+$SO_2$ reaction if the lower rate coefficient from Medeiros et al. (2018) is used.

The Table 1 was modified as follows (changes are shown in red):

**Table 1.** Comparison of observed [$H_2SO_4$] with calculations assuming $H_2SO_4$ formation via oxidation of $SO_2$ by OH and SCIs. [OH] and [$H_2SO_4$] are observed concentrations, [$H_2SO_4$]$^{OH}$ and [$H_2SO_4$]$^{SCI}$ are calculated $H_2SO_4$ produced by oxidation of $SO_2$ by OH and SCI, respectively. Values in square brackets correspond to [$H_2SO_4$]$^{OH}$ concentrations calculated with a rate coefficient for the reaction of OH with $SO_2$ from Medeiros et al. (2018) (see discussion in Sect. 4.2).

| | Daytime: 7:00 – 20:00 | | Night-time: 20:00 – 7:00 | |
|---|---|---|---|---|
| | Median (inter-quartile range) | Mean ± 1σ | Median (inter-quartile range) | Mean ± 1σ |
| [OH], $10^5$ cm$^{-3}$ | 31 (18; 42) | 31 ± 17 | 1.1 (-0.7; 3.0) | 1.7 ± 4.0 |
| [$H_2SO_4$], $10^5$ cm$^{-3}$ | 47 (28; 86) | 63 ± 49 | 4.2 (3.1; 6.4) | 5.8 ± 4.8 |
| | | | | |
| [$H_2SO_4$]$^{OH}$ = a + b×[$H_2SO_4$] | a=(-2.0±0.5)×$10^5$ ; b=0.85±0.02 [a=(-1.1±0.3)×$10^5$ ; b=0.52±0.01] | | a=(-3.1±0.3)×$10^5$ ; b=0.97±0.1 [a=(-1.7±0.2)×$10^5$ ; b=0.60±0.05] | |
| | | | | |
| [$H_2SO_4$]$^{OH}$/ [$H_2SO_4$], % | 95 (79; 129) [58 (48; 78)] | 86 ± 4 [52 ± 2] | 39 (-8; 84) [23 (-4; 52)] | 9 ± 15 [5 ± 9] |
| | | | | |
| 1-[$H_2SO_4$]$^{OH}$/ [$H_2SO_4$], % | 5 (-29; 21) [42 (22; 52)] | 14 ± 4 [48 ± 2] | 61 (16; 108) [77 (48; 104)] | 91 ± 15 [95 ± 9] |
| [$H_2SO_4$]$^{SCI}$/ [$H_2SO_4$], % | 10 (7; 16) | 12 ± 6 | 30 (22; 48) | 38 ± 24 |
| | | | | |
| [$H_2SO_4$]-[$H_2SO_4$]$^{OH}$, $10^5$ cm$^{-3}$ | 1.2 (-12.5; 8.1) [15.2 (8.3; 36.5)] | 4.6 ± 3.2 [11.2 ± 3.4] | 3.0 (0.8; 5.2) [3.2 (1.8; 5.2)] | 3.1± 0.4 [3.3 ± 0.3] |
| [$H_2SO_4$]$^{SCI}$, $10^5$ cm$^{-3}$ | 6.0 (3.7; 8.6) | 6.4 ± 3.7 | 1.4 (1.1; 2.4) | 1.8 ± 1.2 |

Also, we have added some comments to the Sect. 4.2, where this issue is discussed:

"In a recent study of Blitz et al. (2017a, 2017b) a significantly lower rate constant of 5.8×$10^{-13}$ cm$^3$ molecule$^{-1}$ s$^{-1}$ was derived from experiments with vibrationally excited OH (v=1,2,3)+$SO_2$ and using the master equation analysis of the pressure and temperature dependence of their own and some others experimental OH+$SO_2$ reaction rate constants. An even lower rate constant of 4.8×$10^{-13}$ cm$^3$ molecule$^{-1}$ s$^{-1}$ has been derived by Medeiros et al. (2018) applying more detailed master equation analysis of experimental data from Blitz et al. (2017a, 2017b) and some other data. These recent results have not been confirmed by other studies. Also, they have been discussed but not recommended by the latest JPL evaluation (Burkholder et al., 2019).

Using the lower rate constant from Medeiros et al. (2018) in our study would result in about 2 times reduced $H_2SO_4$ production by oxidation of $SO_2$ by OH and would invoke either significantly larger contribution from an additional $H_2SO_4$ source or a lower $H_2SO_4$ uptake coefficient, of about 0.5 instead of unity. As shown in Table 1, the reaction of OH with $SO_2$ would explain only about 50% and 5% of the observed $H_2SO_4$ production during the day and during the night, respectively."

Sincerely,

Alexandre Kukui